# SEMPPL: PREDICTING PSEUDO-LABELS FOR BETTER CONTRASTIVE REPRESENTATIONS

**Matko Bošnjak, Pierre H. Richemond, Nenad Tomasev, Florian Strub, Jacob C. Walker**
**Felix Hill, Lars Holger Buesing, Razvan Pascanu, Charles Blundell, Jovana Mitrovic**
DeepMind
`{matko, richemond, mitrovic}@deepmind.com`

## ABSTRACT

Learning from large amounts of unsupervised data and a small amount of supervision is an important open problem in computer vision. We propose a new semi-supervised learning method, *Semantic Positives via Pseudo-Labels* (SEMPPL), that combines labelled and unlabelled data to learn informative representations. Our method extends self-supervised contrastive learning—where representations are shaped by distinguishing whether two samples represent the same underlying datum (positives) or not (negatives)—with a novel approach to selecting positives. To enrich the set of positives, we leverage the few existing ground-truth labels to predict the missing ones through a $k$-nearest neighbours classifier by using the learned embeddings of the labelled data. We thus extend the set of positives with datapoints having the same pseudo-label and call these *semantic positives*. We jointly learn the representation and predict bootstrapped pseudo-labels. This creates a reinforcing cycle. Strong initial representations enable better pseudo-label predictions which then improve the selection of semantic positives and lead to even better representations. SEMPPL outperforms competing semi-supervised methods setting new state-of-the-art performance of 68.5% and 76% top-1 accuracy when using a ResNet-50 and training on 1% and 10% of labels on ImageNet, respectively. Furthermore, when using selective kernels, SEMPPL significantly outperforms previous state-of-the-art achieving 72.3% and 78.3% top-1 accuracy on ImageNet with 1% and 10% labels, respectively, which improves absolute +7.8% and +6.2% over previous work. SEMPPL also exhibits state-of-the-art performance over larger ResNet models as well as strong robustness, out-of-distribution and transfer performance. We release the checkpoints and the evaluation code at `https://github.com/deepmind/semppl`.

## 1 INTRODUCTION

In recent years, self-supervised learning has made significant strides in learning useful visual features from large unlabelled datasets [Oord et al., 2018; Chen et al., 2020a; Mitrovic et al., 2021; Grill et al., 2020; Caron et al., 2021]. Moreover, self-supervised representations have matched the performance of historical supervised baselines on the ImageNet-1k benchmark [Russakovsky et al., 2015] in like-for-like comparisons as well as outperformed supervised learning in many transfer settings [Tomasev et al., 2022]. While such results show exciting progress in the field, in many real-wold applications often there exists a small amount of ground-truth labelled datapoints making the problem of representation learning semi-supervised.

In this work we propose a novel approach to semi-supervised learning called *Semantic Positives via Pseudo-Labels* (SEMPPL) which incorporates supervised information during the representation learning stage within a self-supervised loss. Unlike previous work which uses the available supervision as targets within a cross-entropy objective, we propose to use the supervised information to help inform which points should have similar representations. We propose to learn representations using a contrastive approach, i.e. we learn the representation of a datapoint (anchor) by maximizing the similarity of the embedding of that datapoint with a set of similar points (positives), while simultaneously minimizing the similarity of that embedding with a set of dissimilar points (negatives). As such, the appropriate construction of these sets of positives and negatives is crucial to the success of

contrastive learning methods. While strategies for sampling negatives have been extensively studied in the literature [Schroff et al., 2015; Harwood et al., 2017; Ge et al., 2018; Wang et al., 2019a; He et al., 2020; Chen et al., 2020c], the sampling of positives has received far less attention.

We propose a novel approach to selecting positives which leverages supervised information. Specifically, we propose using the small amount of available ground-truth labels in order to non-parametrically predict the missing labels (*pseudo-labels*) for the unlabelled data. Note that many previous semi-supervised approaches use pseudo-labels as targets within a cross-entropy-based objective [Van Engelen & Hoos, 2020; Yang et al., 2021]. In SEMPPL we use pseudo-labels in a very different way, i.e. we use them to select positives based on whether two datapoints (we call these *semantic positives*) share the same (pseudo-)label. By maximizing the similarity of a datapoint with its semantic positives we expect to learn representations that are more semantically aligned and as a consequence encode more abstract, higher-level features which should generalise better. To predict informative pseudo-labels, we compare the representations of the unlabelled data with those of the labelled subset and use a $k$-nearest neighbours ($k$-NN) classifier to impute the missing labels.

We simultaneously learn the representation, predict pseudo-labels and select semantic positives. This creates a *virtuous cycle*: better representations enable better pseudo-label prediction which in turn enables better selection of semantic positives and thus helps us learn better representations. Importantly, as the prediction of pseudo-labels and selection of semantic positives does not depend on the exact form of the contrastive objective employed, SEMPPL is compatible with and complements all contrastive losses, e.g. [Chen et al., 2020a;b; Caron et al., 2020; He et al., 2020; Mitrovic et al., 2021] and may even be extended to non-contrastive losses [Grill et al., 2020; Chen & He, 2021].

We evaluate the representations learned with SEMPPL across a varied set of tasks and datasets. In particular, SEMPPL sets new state-of-the-art in semi-supervised learning on ImageNet with $1\%$ and $10\%$ of labels on the standard ResNet-50 ($1\times$) architecture with respectively $68.5\%$ and $76.0\%$ top-1 performance and across larger architectures. When combined with Selective Kernels [Li et al., 2019b], we achieve $72.3\%$ and $78.3\%$ top-1 performance with $1\%$ and $10\%$ labels, respectively, significantly outperforming previous state-of-the-art by absolute $+7.8\%$ and $+6.2\%$ in top-1 performance. We also outperform previous state-of-the-art on robustness and out-of-distribution (OOD) generalisation benchmarks while retaining competitive performance in transfer learning.

Our main contributions are:

- We extend contrastive learning to the semi-supervised setting by introducing the idea of estimating pseudo-labels for selecting semantic positives as a key component especially in the low-label regime,
- We propose a novel semi-supervised method SEMPPL that jointly estimates pseudo-labels, selects semantic positives and learns representations which creates a virtuous cycle and enables us to learn more informative representations,
- We extensively evaluate SEMPPL and achieve a new state-of-the-art in semi-supervised learning, robustness and out-of-distribution generalisation, and competitive performance in transfer.

## 2 SEMANTIC POSITIVES VIA PSEUDO-LABELS

The selection of appropriate positive and negative examples are the cornerstone of contrastive learning. Though the research community has mainly focused on the selection of negatives, positives are equally important as they play a vital role in learning semantic similarity. We thus leverage labelled information as it encodes semantic information to improve the selection of informative positives. Specifically, we expand a self-supervised model to use this labelled data to non-parametrically predict pseudo-labels for the remaining unlabelled data. Using both ground-truth labels and the predicted pseudo-labels, we expand the set of positives with semantic positives.

**Notations** Let $\mathcal{D} = \mathcal{D}_l \cup \mathcal{D}_u$ be a dataset consisting of labelled training data $\mathcal{D}_l = \{(x_i, y_i)\}_{i=1}^N$ and unlabelled training data $\mathcal{D}_u = \{(x_j)\}_{j=N+1}^M$ with $M \gg N$. Let $\mathcal{B}$ be a batch of data of size $B$ with $\mathcal{B} = \{(x_i, y_i)\}_{i=1}^b \cup \{x_j\}_{j=b+1}^B$ where $(x_i, y_i) \in \mathcal{D}_l$ and $x_j \in \mathcal{D}_u$, where the indices $i$, $j$ and $m$ to denote labelled, unlabelled, and all datapoints, respectively. Following established self-supervised learning practices [Chen et al., 2020a;b; Caron et al., 2020; Mitrovic et al., 2021; Dwibedi et al., 2021; Tomasev et al., 2022], we create different views of the data by applying pairs of randomly

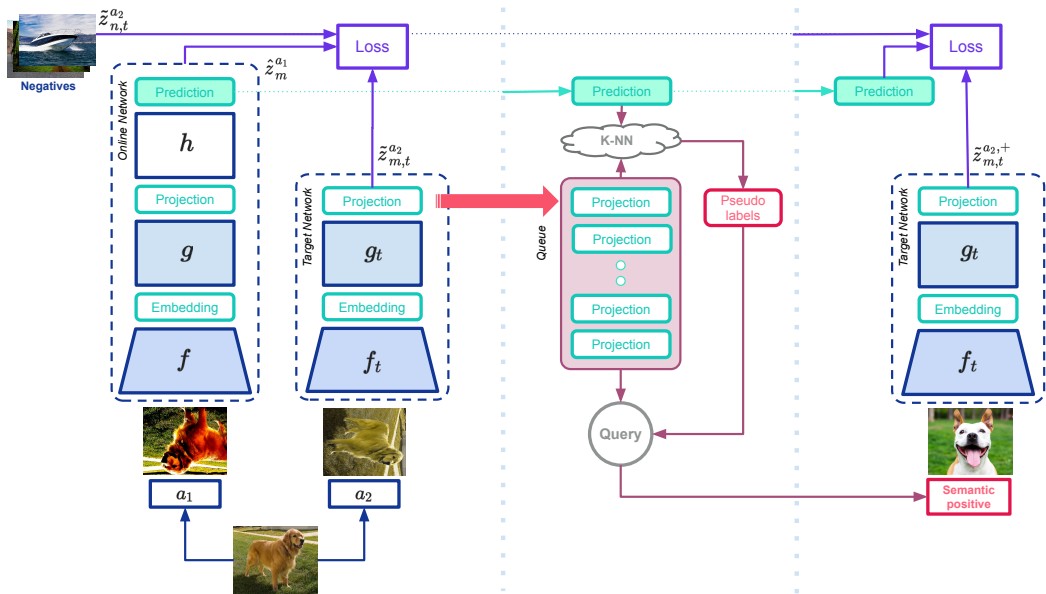

Figure 1: Sketch of SEMPPL. (Left) Standard contrastive pipelines. (Middle) Unlabelled data are tagged with pseudo-labels by using a $k$-NN over projected labelled data. (Right) Semantic positives are queried from the queue and processed to compute an additional contrastive loss.

sampled augmentations $a_1, a_2 \sim \mathcal{A}$ from the augmentation distribution $\mathcal{A}$ proposed in Chen et al. [2020a]. For every datapoint $x_m \in \mathcal{D}$ we denote the corresponding augmentations as $x_m^{a_1}, x_m^{a_2}$.

**Augmentation positives**  We embed one data view $x_m^{a_1}$ via an *online* encoder network $f$ and embed the other data view $x_m^{a_2}$ with a *target* encoder network $f_t$, i.e. we get latent representations $z_m^{a_1} = f(x_m^{a_1})$ and $z_{m,t}^{a_2} = f_t(x_m^{a_2})$. Note that the weights of $f_t$ are an exponential moving average of the weights of $f$. Next, we pass these latent representations through projection and prediction multi-layer perceptrons. Specifically, we use an online projector $g$ and target projector $g_t$, as well as an online predictor $h$, to further transform $z_m^{a_1}$ and $z_{m,t}^{a_2}$; again, the weights of $g_t$ are an exponential moving average of the weights of $g$. We then get $\hat{z}_m^{a_1} = h(g(z_m^{a_1}))$ and $\tilde{z}_{m,t}^{a_2} = g_t(z_{m,t}^{a_2})$ and $l_2$-normalise these; we use $\hat{z}_m^{a_1}, \tilde{z}_{m,t}^{a_2}$ onward as the normalised latent embeddings.

In order to learn the representation of $\hat{z}_m^{a_1}$, we contrast it against the augmentation-based positive $\tilde{z}_{m,t}^{a_2}$ as well as against negatives. For this, we use the contrastive loss:

$$\mathcal{L}_{\text{AUGM}} = -\sum_{m=1}^{B} \log \frac{\varphi(\hat{z}_m^{a_1}, \tilde{z}_{m,t}^{a_2})}{\varphi(\hat{z}_m^{a_1}, \tilde{z}_{m,t}^{a_2}) + \sum_{x_n \in \mathcal{N}(x_m)} \varphi(\hat{z}_m^{a_1}, \tilde{z}_{n,t}^{a_2})} \tag{1}$$

where $\mathcal{N}(x_k)$ is the set of negatives, randomly uniformly sampled from the current batch, $\tilde{z}_{n,t}^{a_2} = g_t(f_t(x_n))$ the target network projection of the negative sample; $\varphi(x_1, x_2) = \tau \cdot \exp(\langle x_1, x_2 \rangle / \tau)$ is the scoring function, $\tau > 0$ is a scalar temperature, and $\langle \cdot, \cdot \rangle$ denotes the Euclidean dot product. Since the representations we contrast are $l_2$-normalised, the dot product effectively turns into cosine similarity.

**Pseudo-label prediction and semantic positives**  Since we have access to a small labelled dataset, we can use the label information to select more informative positives beyond just augmentations of the original image. Specifically, we can associate images with the same label as positives and we call these *semantic positives*. We want to select semantic positives for all the data, not just the labelled subset. For this purpose, we propose to compute pseudo-labels for the unlabelled data and use this to select semantic positives. To compute pseudo-labels we compare the current latent embeddings of the unlabelled data to those of the labelled data. Specifically, we propose to use a first-in-first-out queue $Q$ with capacity $C$ for storing labelled embeddings which we use for computing the pseudo-labels. At the start of training, we simply initialise the queue with random vectors, and use the queue

from the first step. For each batch $\mathcal{B}$, we add the target projection of only the labelled data to the queue, i.e. $Q \leftarrow (\tilde{z}_{i,t}^{a_2}, y_i)$. To predict a pseudo-label for an unlabelled datapoint $x_j$, we first compute the online predictor output $\hat{z}_j^{a_1}$, before retrieving its $k$-nearest neighbours $\{(\tilde{z}_{s,t}^{a_2}, y_s)\}_{s=1}^k$ in cosine similarity from the queue $Q$.[1] Finally, we compute the pseudo-label $\bar{y}_j$ of $x_j$ as:

$$\bar{y}_j = \operatorname*{mode}_{y_s}\{(\tilde{z}_{s,t}^{a_2}, y_s)\}_{s=1}^k \tag{2}$$

where $\operatorname{mode}$ is the mode of the set, tasked with obtaining the most frequent class in the k-nearest neighbours. We use the ground-truth labels (for the labelled data) or the computed pseudo-labels (for the unlabelled data) to select *semantic positives* for every datapoint in $\mathcal{B}$. For each $x_m \in \mathcal{B}$, we uniformly sample over all the embeddings in $Q$ that share the same (pseudo-) label as $x_m$ to get a semantic positive $\tilde{z}_{m,t}^{a_2,+} \sim U(\{(\tilde{z}_{l,t}^{a_2}, y_l) \in Q \mid y_l = pl(x_m)\})$, where $pl(x_m) = y_m$ if $x_m$ is labelled and $pl(x_m) = \bar{y}_m$ if $x_m$ is unlabelled. Next, we include these semantic positives within our representation learning process through the contrastive objective

$$\mathcal{L}_{\text{SEMPOS}} = -\sum_{m=1}^B \log \frac{\varphi(\hat{z}_m^{a_1}, \tilde{z}_{m,t}^{a_2,+})}{\varphi(\hat{z}_m^{a_1}, \tilde{z}_{m,t}^{a_2,+}) + \sum_{x_n \in \mathcal{N}(x_m)} \varphi(\hat{z}_m^{a_1}, \tilde{z}_{n,t}^{a_2})} \tag{3}$$

Taking these two losses (1) and (3) together, we propose to learn representations in our method SemPPL by minimising the following total loss

$$\mathcal{L}_{\text{SEMPPL}} = \mathcal{L}_{\text{AUGM}} + \alpha \mathcal{L}_{\text{SEMPOS}} \tag{4}$$

where $\alpha$ controls the ratio between these sub-losses.

## 2.1 IMPLEMENTATION DETAILS

**Architecture** We use Residual Networks [He et al., 2016] (v1; pre-activation as customary in the literature) for $f$ and $f_t$ and use either 50 or 200 layers deep networks and with a width multiplier ranging from $1\times$ to $4\times$. As in [Grill et al., 2020; Tomasev et al., 2022], we use multi-layer perceptrons with 2 layers of size 4096 and 256, with batch normalisation [Ioffe & Szegedy, 2015] and rectified linear activation.

**Self-supervised learning method** We use RELICv2 [Tomasev et al., 2022] as our default self-supervised training objective due to its competitive performance. Therefore, we add an invariance penalty on top of Equation 4 to further enforce the similarity constraints and regularize the learning process as detailed in Appendix B. We also explore other self-supervised learning objectives in Section 4.

**Algorithm parameters** We use a queue of capacity $C = 20B$, with batch size $B = 4096$, and temperature $\tau = 0.2$ while randomly sampling negatives from the current batch; we take $|\mathcal{N}(x)| = 10$ negatives in total. For augmentations, we use the standard SIMCLR augmentations [Chen et al., 2020a] and the RELICv2 multi-crop and saliency-based masking [Tomasev et al., 2022]; we use 4 large views and 2 small views for augmentation positives and 3 semantic positives. The semantic positives are computed with a $k$-NN with $k = 1$ (see the analysis section in Appendix D); we build a single $k$-NN instance per augmentation $a$ queried with all the augmentations where $|a| = 4$. This produces $|a|^2 = 16$ $k$-NN induced pseudo-labels in total for each unlabelled image among which we then perform majority voting to compute the final pseudo-label.

**Optimisation** Our networks are optimized with LARS [You et al., 2017]. Our base learning rate is 0.3 and we train our models for 300 epochs with a learning rate warm-up period of 10 epochs and cosine decay schedule thereafter. We use a weight decay of $10^{-6}$ and batch size $B = 4096$. We exclude the biases and batch normalisation parameters both from LARS adaptation and weight decay. The exponential moving average parameter for target networks is 0.996. Our pseudo-code is described in the appendix along with precise architectural and implementation details. Pretrained model checkpoints and code will be made available on GitHub.

---

[1]We use the cosine similarity as the embeddings are normalised.

## 3 EXPERIMENTAL RESULTS

To evaluate SEMPPL, we pre-train representations using 1% and 10% labelled data from the ImageNet dataset [Russakovsky et al., 2015] based on the splits from Chen et al. [2020a]. We then test SEMPPL in semi-supervised classification, robustness and out-of-distribution generalisation tasks. Lastly, we probe the transfer capabilities of the representations to other image classification datasets. For a complete set of results and experimental details, please see the Appendix A.

### 3.1 SEMI-SUPERVISED LEARNING

In Table 1, we report top-1 accuracy on the ImageNet test set when either 1% or 10% of the data is labelled for the ResNet-50 architecture as well as deeper and wider ResNets. SEMPPL achieves top-1 accuracy of 68.5% with 1% of labels, significantly outperforming the previous state-of-the-art SimMatch [Zheng et al., 2022] by an absolute +1.3% in ImageNet test accuracy. With 10% of label data, our top-1 accuracy on ResNet-50 reaches 76.0%, outperforming the previous state-of-the-art PAWS [Assran et al., 2021] in semi-supervised learning. SEMPPL outperforms competing representation learning methods across the board, achieving state-of-the-art performance on all ResNet-50 2×, ResNet-50 4× and , in both the 1% and 10% labelled settings. SEMPPL does not use, and therefore excludes from comparison, distillation from larger networks as in [Chen et al., 2020b; Pham et al., 2021].

Similar to [Chen et al., 2020b], we also tested SEMPPL on ResNets with *Selective Lernels* (SK) [Li et al., 2019b]. This increases the encoder parameter count to 27.5M. We thus achieve a new absolute state-of-the-art of 72.3% and 78.3% top-1 accuracies, respectively, when using 1% and 10% of labelled data. Finally, SEMPPL reaches a new state-of-the-art using 76.0 and 80.5 on 1% and 10% of labels without self-distillation with a ResNet-200 2× + SK architecture.

For implementation details of the semi-supervised results and additional results, see the Appendix A.1.

Table 1: Top-1 accuracy (in %) for ResNet encoders with different depth and width.

| | ResNet-50 1× | | ResNet-50 2× | | ResNet-50 4× | | ResNet-200 2× | |
|---|---|---|---|---|---|---|---|---|
| Method | Top-1 | | Top-1 | | Top-1 | | Top-1 | |
| | 1% | 10% | 1% | 10% | 1% | 10% | 1% | 10% |
| SimCLR [Chen et al., 2020a] | 48.3 | 65.6 | 58.5 | 71.7 | 63.0 | 74.4 | - | - |
| BYOL [Grill et al., 2020] | 53.2 | 68.8 | 62.2 | 73.5 | 69.1 | 75.7 | 71.2 | 77.7 |
| RELICv2 [Tomasev et al., 2022] | 58.1 | 72.4 | 64.7 | 73.7 | 69.5 | 74.6 | 72.1 | 76.4 |
| SimCLRv2 [Chen et al., 2020b] | 57.9 | 68.4 | 66.3 | 73.9 | - | - | - | - |
| CoMatch [Li et al., 2021a] | 66.0 | 73.7 | - | - | - | - | - | - |
| PAWS [Assran et al., 2021] | 66.5 | 75.5 | 69.6 | 77.8 | 69.9 | 79.0 | - | - |
| SimMatch [Zheng et al., 2022] | 67.2 | 74.4 | - | - | - | - | - | - |
| SemPPL (ours) | **68.5** | **76.0** | **71.9** | **78.6** | **72.5** | **79.3** | **74.8** | **80.4** |
| SimCLRv2 + SK [Chen et al., 2020b] | 64.5 | 72.1 | 70.6 | 77.0 | - | - | - | - |
| SemPPL + SK (ours) | **72.3** | **78.3** | **74.5** | **79.8** | - | - | **76.0** | **80.5** |

### 3.2 ROBUSTNESS AND OOD GENERALISATION

We evaluate the robustness and generalisation abilities of SEMPPL on ImageNetV2 [Recht et al., 2019], ImageNet-C [Hendrycks & Dietterich, 2019], ImageNet-R [Hendrycks et al., 2021] and ObjectNet [Barbu et al., 2019] which have all been purposefully constructed to test different robustness and generalisation aspects. We evaluate all three variants on ImageNetV2: matched frequency (MF), Threshold 0.7 (T-0.7) and Top Images (TI). When evaluating PAWS, we used the publicly available checkpoints. Table 2 shows good robustness and generalisation ability of the representations learned with SEMPPL. SEMPPL sets the new state-of-the-art performance (outperforming even the supervised baseline) on 4 out of 5 datasets, while outperforming PAWS across all datasets. SEMPPL also outperforms SimMatch on 4 out of 5 datasets. For more details on the evaluation protocols and results for ImageNet-C see the Appendix A.2.

Table 2: Top-1 accuracy (in %) for ImageNetV2, ImageNet-R and ObjectNet.

| Method | | Robustness | | | OOD generalization | |
|---|---|---|---|---|---|---|
| | | MF | T-0.7 | Ti | ImageNet-R | ObjectNet |
| Supervised (100% labels) [Lim et al., 2019] | | 65.1 | 73.9 | 78.4 | 24.0 | **26.6** |
| *Semi-supervised (10% labels)* | | | | | | |
| PAWS [Assran et al., 2021] | | 64.5 | 73.7 | 78.9 | 23.5 | 23.8 |
| SimMatch [Zheng et al., 2022] | | 63.8 | 73.2 | 78.3 | **25.0** | 24.5 |
| SemPPL (ours) | | **65.4** | **74.1** | **79.6** | 24.4 | **25.3** |

Table 3: Top-1 accuracy (in %) on the full suite of transfer tasks.

| Method | Food101 | CIFAR10 | CIFAR100 | Birdsnap | SUN397 | Cars | Aircraft | DTD | Pets | Caltech101 | Flowers |
|---|---|---|---|---|---|---|---|---|---|---|---|
| Supervised-IN [Chen et al., 2020a] | 72.3 | **93.6** | 78.3 | 53.7 | 61.9 | 66.7 | 61.0 | 74.9 | 91.5 | **94.5** | 94.7 |
| *Semi-supervised (10% labels)* | | | | | | | | | | | |
| PAWS [Assran et al., 2021] | 79.1 | 92.3 | 76.3 | 62.0 | 66.1 | **75.7** | 61.4 | 77.0 | 92.2 | 91.9 | **96.5** |
| SimMatch [Zheng et al., 2022] | 71.7 | **93.6** | **78.4** | – | – | 69.7 | – | 75.1 | **92.8** | – | 93.2 |
| SEMPPL (ours) | **80.2** | 92.5 | 77.6 | **64.2** | **66.3** | 75.5 | **63.9** | **77.8** | 92.5 | 93.0 | 96.3 |

## 3.3 TRANSFER LEARNING

We evaluate the generality of SEMPPL representations by testing whether the features learned on ImageNet are useful across different datasets and tasks. Specifically, we evaluate the transfer performance of SEMPPL on a set of 11 image classification datasets commonly used in the contrastive literature under the linear protocol [Grill et al., 2020; Chen et al., 2020a; Dwibedi et al., 2021; Mitrovic et al., 2021; Tomasev et al., 2022]. For the linear protocol, the pretrained encoder is frozen and a randomly initialized linear classifier is trained on top using the training data from the target dataset. We report standard metrics for each dataset as well as performance on a held-out test set. For more details on the evaluation protocol see the Appendix A.3. Table 3 compares the transfer performance of representations pretrained using the supervised baseline [Chen et al., 2020a], PAWS [Assran et al., 2021], SimMatch [Zheng et al., 2022] and our method SEMPPL. SEMPPL outperforms the supervised baseline on 8 out of 11 datasets, PAWS on 9 out of 11 datasets, while showing competitive performance to SimMatch, outperforming it on 4 out of 7 datasets.

## 3.4 FULL LABELLED DATASET

We also assess how SEMPPL behaves in a fully supervised setting. For this purpose, we select semantic positives based on the ground-truth labels and fine-tune the learned representations with the full ImageNet dataset. We compare against strong supervised baselines on ResNets as well as against recent performant network architectures that are extensions of the ResNet, e.g. [Liu et al., 2021b; 2022]. Our method reaches 79.7% top-1 accuracy on a ResNet 50 outperforming a number of strong supervised baselines. When we add selective kernels to a ResNet 50, we achieve 82% top-1 accuracy outperforming recent transformers architecture

Figure 2: Top-1 accuracy for ResNet50 with 100% of the labels across augmentations, initializations and networks.

| Method | Params | Top-1 |
|---|---|---|
| *Supervised (ResNet-50)* | | |
| + AutoAugment [Cubuk et al., 2019] | 27M | 77.6 |
| + MaxUp [Gong et al., 2021] | 27M | 78.9 |
| *Representation Learning (ResNet-50)* | | |
| SEMPPL (SimCLR base) | 27M | 76.0 |
| SEMPPL (BYOL base) | 27M | 77.7 |
| SEMPPL (ReLICv2 base; ours) | 27M | 79.7 |
| *Other Architectures* | | |
| Swin-T [Liu et al., 2021b] | 29M | 81.3 |
| ConvNeXt [Liu et al., 2022] | 29M | 82.1 |
| SEMPPL + SK (ours) | 29M | 82.0 |

ture [Liu et al., 2021b], and matching highly tuned ConvNext [Liu et al., 2022]. Therefore, SEMPPL may also be considered as a promising pretraining method in the supervised learning setting.

## 4 ANALYSIS

We analyse the impact of different design choices in SEMPPL on downstream performance. In this section, we focus the behaviour and impact of pseudo-labels and semantic positives on learning representations. For further analyses and experimental details, please see Appendix D.

**Semantic positives across self-supervised learning objectives**  With SEMPPL we extend the set of positives to include semantic positives based on predicted pseudo-labels; we can combine these ideas with other self-supervised methods. In Table 4a, we additionally evaluate SEMPPL on the non-contrastive self-supervised method BYOL [Grill et al., 2020]. BYOL replaces the contrastive loss in Equation 4 with an $l_2$ loss. Importantly, we follow the training pipeline (e.g. augmentation, hyper-parameters etc.) from [Grill et al., 2020] to fairly highlight the impact of SEMPPL. We observe a drastic improvement when adding semantic positives. With 1% labels on ImageNet BYOL improves by absolute +3.9% and by absolute +3.6% when using 10% labels. For completeness we have also highlighted the contribution of SEMPPL when using RELICv2 as the base self-supervised objective which is our default implementation. For 1% labeled data, we see an absolute improvement of +10.4% in top-1 accuracy, while for 10% labels we see a gain of absolute +3.6% in top-1 accuracy. In summary, we see that SEMPPL can be easily combined with other self-supervised objectives to yield significant improvements and can be used a plug-and-play module in semi-supervised learning.

**The contribution of pseudo-labels and semantic positives**  We examine the impact of omitting pseudo-label prediction and semantic positives from learning representations. Specifically, we ablate the use of pseudo-labels when selecting semantic positives for unlabelled datapoints, i.e. we only use labelled images when retrieving semantic positives. In Table 4b (middle row), removing pseudo-label prediction significantly decreases performance both in the 1% and 10% label settings. In addition, the low-label regime (1% labels) suffers a stronger performance decrease −6.6% than the 10% labels regime, −4.9%. This underscores the importance of pseudo-label estimation and subsequent selection of semantic positives for unlabelled data especially in the low-data regime. Going a step further, we remove semantic positives even for the labelled data, falling back to vanilla RELICv2. In Table 4b (bottom row), we see again a significant drop in performance for both the 1% and 10% label settings with a sharper drop for the low-label regime. Together these highlights the importance of both including semantic positives for labelled data as well as using pseudo-label prediction for selecting semantic positives for unlabelled data in order to learn informative representations in a semi-supervised setting in a label-efficient way.

**Precision and Recall of pseudo-labels.**  In Figure 3, we analyse the behaviour of pseudo-labels by looking at the precision and recall as training progresses. We train a ResNet-50 for 100 epochs using 10% labels with SEMPPL on ImageNet. As we have 4 large views there will be in total 16 votes cast and then the pseudo-label will be estimated using majority voting. We want to measure how often these 16 votes agree or disagree; we denote as voting threshold the number $k$ where at least $k$ votes have been cast for one class. We see that as training progresses the precision across all thresholds increases as expected. This means that the pseudo-label prediction is bootstrapping itself to become more accurate, which enables us to select better semantic positives and thus learn more informative representations as training progresses, i.e. we have a virtuous cycle of representation learning and pseudo-label prediction. Furthermore, precision is an increasing function of the voting

Table 4: Top-1 test accuracy (in %) with a ResNet50 pretrained on ImageNet with 1% and 10% labels.

|  | Top-1 | |
|---|---|---|
|  | 1% | 10% |
| BYOL [Grill et al., 2020] | 53.2 | 68.8 |
| SEMPPL with BYOL | 57.1 | 72.4 |
| ReLICv2 [Tomasev et al., 2022] | 58.1 | 72.4 |
| SEMPPL with RELICv2 (ours) | **68.5** | **76.0** |

(a) Trained on a different self-supervised objective.

|  | 1% labels | 10% labels |
|---|---|---|
| SEMPPL | **68.5** | **76.0** |
| - Pseudo-labels | 61.9 | 71.1 |
| - Semantic Positives | 58.1 | 72.4 |

(b) Removing pseudo-labelling and semantic positives in SEMPPL.

Table 5: Top-1 test accuracy (in %) with a ResNet50 pretrained on ImageNet 10% labels for 100 epoches when using ground truth labels instead of pseudo-labels while retrieving semantic positives (oracle); PL accuracy holds for pseudo-label accuracy.

| 10% labels | Top-1 | PL accuracy |
|---|---|---|
| SEMPPL | 69.9 | 69.7 |
| SEMPPL (+oracle) | **71.6** | **76.9** |

threshold throughout training and is highest for the biggest voting threshold. This indicates how confident we can be in the accuracy of pseudo-label prediction, and thus how confident we can be that an appropriate semantic positive has been selected. Yet, we see that the recall for individual thresholds is also increasing as training progresses but that the recall decreases as we increase the voting threshold. This is expected as there is always a trade-off between precision and recall.

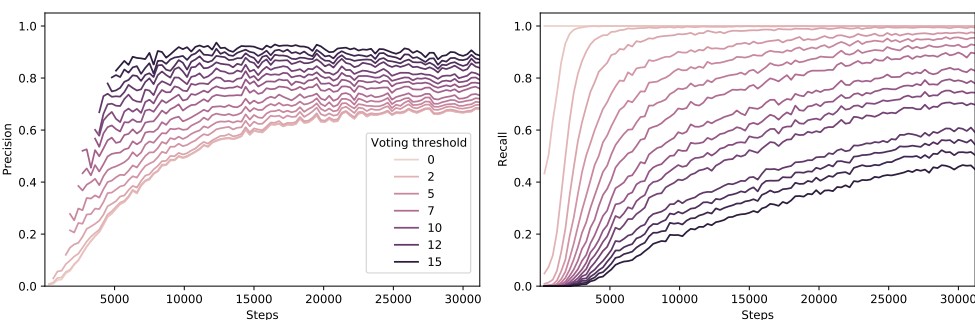

Figure 3: Precision and recall for pseudo-labels computed based on $k$-nearest neighbours when trained on ImageNet with 10% labels over 100 epoches.

**Noise in pseudo-label prediction**   In Figure 3, we observe that the proportion of correctly predicted pseudo-labels at the end of training is reasonably high ($60\%$ accuracy of voting threshold 0). Yet, it also means that $40\%$ of pseudo labels are still incorrectly predicted. As the incorrect prediction results in suboptimal semantic positives selection, i.e., SEMPPL does not select semantic positives from the same class as the datapoint, this behavior may ultimately worsen the quality of extracted features. To quantify this phenomenon, we train the representation with SEMPPL where an oracle replaces the pseudo-label prediction with ground-truth labels and those are used for selecting semantic positives. In Table 5, we train the representations for 100 epochs on ImageNet with $10\%$ labels. There, the oracle increases the top-1 performance of $1.7\%$ in the test set with $10\%$. Besides, the pseudo-label accuracy also gets $6.2\%$ higher. It thus confirms that incorrect pseudo-label predictions, and incorrect semantic positives retrieval, hurts learning informative representations and the downstream performance. Yet, the oracle performance remains close to the actual performance of SEMPPL, illustrating the method's robustness.

Further ablations on other design choices, such as the number of semantic positives, the use of view voting, the choice of $k$ in $k$-NN, queue length and training duration can be found in Appendix D.2.

## 5   RELATED WORK

**Semi-supervised learning**   In the semi-supervised regime [Cheplygina et al., 2019; Van Engelen & Hoos, 2020; Yang et al., 2021; Alizadehsani et al., 2021], one can either pre-train a model on unlabelled data and subsequently fine-tune it on labelled data, or train both jointly. Joint training on labelled and unlabelled data often involves combining the two losses [Grandvalet & Bengio, 2004; Miyato et al., 2018; Zhai et al., 2019; Verma et al., 2019; Berman et al., 2019; Xie et al., 2020a]. Pseudo-label self-training approaches [Zoph et al., 2020] present an important alternative, first inferring approximate pseudo-labels for the unlabelled examples, and then incorporating them in supervised losses. Pseudo-labels can either be generated prior to a subsequent supervised learning

phase [Yarowsky, 1995; Riloff, 1996; Lee et al., 2013] or jointly in an online fashion [Berthelot et al., 2019; 2020; Sohn et al., 2020]. These methods may benefit from pseudo-label confidence measures [Sohn et al., 2020; Rizve et al., 2021; Zhang et al., 2021] as well as thresholding [Xu et al., 2021], temporal ensembling [Laine & Aila, 2017], or stronger regularization to mitigate bias in early model training [Sajjadi et al., 2016; Arazo et al., 2020]. The use of pseudo-labels with rebalancing has shown improvements, both in class-imbalanced problems [Wei et al., 2021] and in a general context [Wang et al., 2022]. Teacher-student network configurations for generating and utilising pseudo-labels have also shown promise [Tarvainen & Valpola, 2017; Luo et al., 2018; Ke et al., 2019; Xie et al., 2020b; Cai et al., 2021; Pham et al., 2021]. Co-training uses different feature extractors for different data views and alternates between pseudo-labelling and training phases [Blum & Mitchell, 1998; Qiao et al., 2018]. Good performance has been reached by using consistency losses between pseudo-labels of different inputs [Verma et al., 2019; Hu et al., 2021].

Predicting view assignments with support samples [Assran et al., 2021] (PAWS) has resulted in substantial performance improvements, with the idea that the assigned pseudo-labels ought to be similar across multiple views of the same image. Recent work has shown that incorporating label information in positive selection in contrastive methods is highly promising, compared to the cross-entropy loss in the fully supervised case [Khosla et al., 2020]. Our method demonstrates a similar utility of pseudo-labels for semi-supervised problems, and differs from competing ones in the following ways. Unlike DebiasPL [Wang et al., 2022] that uses an adaptive margin loss, SemPPL does not seek to directly address or re-shape the distribution of pseudo-labels. Unlike SimCLRv2 [Chen et al., 2020b], we do not rely on self-distillation procedures. In contrast with PAWS [Assran et al., 2021], we fully leverage the contrastive approach for semi-supervised learning; not using positives only for training means SEMPPL does not require specific care like pseudo-labels sharpening to stabilize learning and avoid representational collapse. SEMPPL is more closely related to CoMatch [Li et al., 2021a] that also uses bootstrapping to improve pseudo-labels representational quality, but is conceptually much simpler, avoiding phases of distributional alignment and of performing graph-based contrastive learning. In a similar vein, SimMatch [Zheng et al., 2022] also uses a memory buffer to propagate pseudo-labels, but has a more complex objective than SEMPPL and equally requires additional phases of pseudo-labels unfolding and aggregation to function.

**Self-supervised learning** Major advances in learning useful representations from unlabelled data [Liu et al., 2021a; Goyal et al., 2021] can be seen as a paradigm shift, since these methods have recently been competitive with supervised training baselines [Tomasev et al., 2022]. A number of self-supervised learning methods involve contrasting multiple views of the data [Oord et al., 2018; Bachman et al., 2019; Chen et al., 2020a; He et al., 2020; Grill et al., 2020; Dwibedi et al., 2021]. Similar performance were also achieved by bootstrapping-based multi-view learning [Grill et al., 2020; Richemond et al., 2020; Chen & He, 2021; Zbontar et al., 2021; Wang et al., 2021], or involving explicit clustering steps [Caron et al., 2020; Asano et al., 2020; Li et al., 2021b]. An explicit causally-motivated invariance loss, when used in conjunction with the contrastive objective, has been shown to lead to more compact representations, and desirable generalisation properties [Mitrovic et al., 2021; Tomasev et al., 2022]. Contrastive approaches are not always used in self-supervised methods [He et al., 2021; Ermolov et al., 2021; Chen et al., 2022]. Transformer-specific methods have been devised [Caron et al., 2021; Chen et al., 2021; Zhai et al., 2022].

## 6 CONCLUSION

In this work, we propose SEMPPL, a novel semi-supervised learning method to incorporate semantic positives in self-supervised objectives by taking advantage of pseudo-labels. Through extensive empirical evaluation, we demonstrated that our approach achieves state-of-the-art semi-supervised performance on ImageNet across several ResNet architectures as well as on the robustness, out-of-distribution generalization and transfer tasks. We also show that SEMPPL can be easily combined with other existing self-supervised methods and is a promising direction to pre-train networks also in a fully supervised learning regime. Our analyses suggest that the role of pseudo-labels in selecting positives for semi-supervised contrastive methods might be underappreciated. Despite widespread use in semi-supervised applications, pseudo-labels are less understood and have been explored far less in the context of self-supervised methods. We hope this study, which shows empirically that prior work has under-utilized pseudo-labels, may help bridge that gap.

REPRODUCIBILITY STATEMENT

We documented the model and experimental evaluation in the main body of the paper and added further details in the appendix. Concretely, we explain implementation details and sweep parameters in Appendix A and Appendix D.1, the invariance loss in Appendix B and give details on data augmentations in Appendix E. The model pseudo-code is in Appendix C. The datasets used in the experiments are freely available from their respective sources. We also open source SemPPL pre-trained checkpoints from our experiments, namely ResNet-50 1×, 2× and 4× as well as ResNet-200 2× together with the evaluation code at https://github.com/deepmind/semppl.

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

# A   ADDITIONAL RESULTS AND IMPLEMENTATION DETAILS

## A.1   SEMI-SUPERVISED DETAILS AND RESULTS

**Implementation details.**   In this work, we follow the protocol of Chen et al. [2020b]; Assran et al. [2021] for fine-tuning from the first layer of the projector and initialize both the encoder and the first layer of the projector with the parameters of the pretrained model. We add the randomly initialized classifier on top of the first layer of the projector (after the non-linearity). We train all the weights (pretrained and classifier weights) using either $1\%$ or $10\%$ of the ImageNet-1k training data, and we use the splits introduced in Chen et al. [2020a] and used in all the methods to compare to Grill et al. [2020]; Caron et al. [2020]; Dwibedi et al. [2021]; Lee et al. [2021]; Mitrovic et al. [2021]; Tomasev et al. [2022]; Assran et al. [2021].

At training time we randomly crop the image, resize it to $224 \times 224$, and then randomly apply a horizontal flip. At test time we resize images to 256 pixels along the shorter side with bicubic resampling and apply a $224 \times 224$ center crop to it. Both at training and testing times we subtract from the color channels the average channel value and divide it by the standard deviation of the channel value (as computed on ImageNet-1k).

We use a cross entropy loss and stochastic gradient descent with Nesterov momentum of $0.9$ to fine-tune the model. For both $1\%$ and $10\%$ settings, we train for 30 epochs and decay the initial learning rate by a factor $0.2$ at 18 and 24 epochs. Following the approach of Caron et al. [2020], we pick different learning rates for the encoder (and the first projector layer) and for the classifier weights. We do not use any weight decay or other regularization techniques. We sweep over batch sizes values in $\{512, 1024, 2048\}$, encoder base learning rate values in $\{0.005, 0.0035, 0.003, 0.0025, 0.002, 0.001\}$, and linear layer base learning rate values in $\{0.5, 0.3, 0.2, 0.1, 0.05, 0.025\}$.

Table 6: Top-1 and Top-5 accuracies (in %), after semi-supervised fine-tuning with a fraction of ImageNet labels, for a ResNet-50 encoder across a number of representation learning methods.

| Method | Top-1 | | Top-5 | |
|---|---|---|---|---|
| | 1% | 10% | 1% | 10% |
| Supervised [Zhai et al., 2019] | 25.4 | 56.4 | 48.4 | 80.4 |
| *Pseudo labels in classification:* | | | | |
| MPL [Pham et al., 2021] | - | 73.9 | - | - |
| *Representation learning methods:* | | | | |
| SimCLRv2 [Chen et al., 2020b] | 57.9 | 68.4 | - | - |
| SimCLRv2 + self distillation [Chen et al., 2020b] | 60.0 | 70.5 | - | - |
| CoMatch [Li et al., 2021a] | 66.0 | 73.7 | 86.4 | 91.6 |
| PAWS [Assran et al., 2021] | 66.5 | 75.5 | - | - |
| DebiasPL [Wang et al., 2022] | 67.1 | - | 85.8 | - |
| SimMatch [Zheng et al., 2022] | 67.2 | 74.4 | 87.1 | 91.6 |
| SEMPPL (ours) | **68.5** | **76.0** | **88.2** | **92.7** |
| SimCLRv2 + Selective Kernels [Li et al., 2019b] | 64.5 | 72.1 | 86.7 | 91.4 |
| SEMPPL (ours) + Selective Kernels | **72.3** | **78.2** | **90.6** | **93.9** |

**Additional results and larger networks.**   When the architecture of the ResNet-50 is modified to include *selective kernels* [Li et al., 2019b], we see significant gains in performance at the expense of additional weights. Our implementation of selective kernels is standard and follows rigorously Li et al. [2019b] for a total of 27.5 million weights instead of of 25.5 million for a regular ResNet-50. Specifically, we use 2 channels, two convolution kernels of $(3, 3)$ and $(5, 5)$ with the latter implemented as a $(3, 3)$ dilated convolution with rate 2, and 32 grouped convolutions. Unlike SimCLRv2 [Chen et al., 2020b], we implement our group convolutions explicitly, and do not use the additional *ResNet-D* architectural modification from He et al. [2019]. When using selective kernels our performance after finetuning with 1% of labels is the same as that of SimCLRv2 after finetuning with 10% of labels.

Additionally, in order to investigate the robustness and scalability of these results, we further test the generality of SEMPPL by learning representations on larger (both deeper and wider) ResNet encoders. Table 1 testifies to SEMPPL outperforming the competing representation learning methods across all the architectures, both in the $1\%$ and the $10\%$ labelled settings. Also, as our flagship result we reach $80.4\%$ top-1 accuracy on ResNet-200 $2\times$ with $10\%$ of ImageNet-1k labels. Just as in the ResNet-50 $1\times$ case this figure is comparable with the fully supervised accuracy attained by historical methods. $80.1\%$ top-1 is defined as in Grill et al. [2020] with standard RandAugment [Cubuk et al., 2020] data augmentation. However it's certainly a few percentage accuracy points away from results obtained with optimal current training protocols [Bello et al., 2021]. We also note that SEMPPL is pre-trained for 300 epochs in all cases. This, rather than the 1000 epochs used as standard by most other representation learning methods, again compares with a typical figure of 200 epochs used in supervised learning. Overall this hints at SEMPPL having achieved close to an order of magnitude gain in label efficiency (compared to supervised learning) at a similar epochs budget.

Our final networks were optimized using tranches of between 128 (for a ResNet-50) and 512 (for the largest ResNets) Cloud TPUv3s all during 300 epochs each irrespective of size. This required around a day of computation time per run and tranche for a ResNet-50 on 128 devices, time which scaled approximately linearly with the number of parameters on larger networks, depending on the actual network.

## A.2 ROBUSTNESS AND OOD GENERALIZATION

We test the robustness and out-of-distribution (OOD) generalization abilities of representations learned via SEMPPL on several detasets. We use ImageNetV2 [Recht et al., 2019] and ImageNet-C [Hendrycks & Dietterich, 2019] datasets to evaluate robustness and the datasets ObjectNet [Barbu et al., 2019] and ImageNet-R [Hendrycks et al., 2021] to evaluate the OOD generalization.

The ImageNetV2 dataset [Recht et al., 2019] has three sets of 10000 images (matched frequency (MF), Threshold 0.7 (T-0.7) and Top Images (TI)) that were collected to have a similar distribution to the ImageNet test set. The ImageNet-C dataset [Hendrycks & Dietterich, 2019] consists of 15 synthetically generated corruptions of 5 different severities (e.g. blur, noise) that are applied to the ImageNet validation set. The ImageNet-R dataset [Hendrycks et al., 2021] consists of 30000 different renditions (e.g. paintings, cartoons) of 200 ImageNet classes; the aim of this dataset is to test the generalization ability to different textures and other naturally occurring style changes that are out-of-distribution to the ImageNet training data. The ObjectNet dataset [Barbu et al., 2019] has 18574 images from differing viewpoints and backgrounds compared to the ImageNet training set.

On all datasets we evaluate the representations learned on a standard ResNet50 encoder under a linear evaluation protocol. We freeze the pretrained representations (no gradient updates) and train a linear classifier on top of the output of the ResNet-50 encoder using the full labelled ImageNet training set. We perform the test evaluation zero-shot, i.e the above datasets are not seen during the training of the representation or classifier.

We provide a detailed breakdown across the different ImageNet-C corruptions in Table 7. Our proposed approach SEMPPL outperforms both the supervised baseline, on 12 out of 15 corruptions, as well as the competing semi-supervised representation learning model PAWS, on 12 out of 15 corruptions (notably, over all Blur, Weather and Digital corruptions).

Table 7: Top-1 accuracies (in %) for OOD generalisation on Gauss, Shot, Impulse, Blur, Weather, and Digital corruption types of ImageNet-C.

| Method | Gauss | Shot | Impulse | Defocus | Glass | Motion | Zoom | Snow | Frost | Fog | Bright | Contrast | Elastic | Pixel | JPEG |
|---|---|---|---|---|---|---|---|---|---|---|---|---|---|---|---|
| | | | | | Blur | | | | Weather | | | | Digital | | |
| Supervised [Lim et al., 2019] | 37.1 | 35.1 | 30.8 | 36.8 | **25.9** | 34.9 | **38.1** | 34.5 | 40.7 | 56.9 | 68.1 | 40.6 | **45.6** | 32.6 | 56.0 |
| *Semi-supervised representations:* | | | | | | | | | | | | | | | |
| PAWS [Assran et al., 2021] | **43.5** | **40.6** | **33.5** | 38.7 | 19.7 | 34.1 | 32.8 | 40.3 | 44.7 | 64.0 | 70.5 | 59.7 | 42.4 | 38.5 | 55.1 |
| SEMPPL(ours) | 41.3 | 39.1 | 30.0 | **41.9** | 23.2 | **37.5** | 34.0 | **40.5** | **45.5** | **64.4** | **71.9** | **60.6** | 44.2 | **45.1** | **57.7** |

## A.3 TRANSFER

To further evaluate the usefulness of the learned representations, we evaluate how well they transfer across datasets. For this, we follow the standard evaluation protocol outlined in Grill et al. [2020];

Chen et al. [2020a]. We evaluate SEMPPL across the linear evaluation protocol which consists of freezing the encoder and only training a randomly initialized linear classifier on top of the encoder. In line with prior work [Chen et al., 2020a; Grill et al., 2020; Dwibedi et al., 2021], we test SEMPPL representations on the following datasets: Food101 [Bossard et al., 2014], CIFAR10 [Krizhevsky et al., 2009], CIFAR100 [Krizhevsky et al., 2009], Birdsnap [Berg et al., 2014], SUN397 (split 1) [Xiao et al., 2010], DTD (split 1) [Cimpoi et al., 2014], Cars [Krause et al., 2013] Aircraft [Maji et al., 2013], Pets [Parkhi et al., 2012], Caltech101 [Fei-Fei et al., 2004], and Flowers [Nilsback & Zisserman, 2008], where we compare the downstream performance of SEMPPL to that of other reported semi-supervised methods on 10% of labels. Across these datasets there are differences in terms of metrics used for selecting the best hyper-parameters as well the reporting of the final results. In line with prior work [Chen et al., 2020a; Grill et al., 2020; Dwibedi et al., 2021], for Food101 [Bossard et al., 2014], CIFAR10 [Krizhevsky et al., 2009], CIFAR100 [Krizhevsky et al., 2009], Birdsnap [Berg et al., 2014], SUN397 (split 1) [Xiao et al., 2010], DTD (split 1) [Cimpoi et al., 2014], and Cars [Krause et al., 2013] we report the Top-1 accuracy on the test set, and for Aircraft [Maji et al., 2013], Pets [Parkhi et al., 2012], Caltech101 [Fei-Fei et al., 2004], and Flowers [Nilsback & Zisserman, 2008] we report the mean per-class accuracy. For DTD and SUN397 we only use the first split of the 10 provided splits in the dataset as per Chen et al. [2020a]; Grill et al. [2020]; Dwibedi et al. [2021].

In these experiments, models are initially trained on the training sets of the individual datasets, and the validation sets are used to select the best hyperparameters from the executed hyperparameter sweeps. Once the best hyperparameters have been selected, the final models are trained on a merged dataset containing both the training and the validation split and evaluated on the held-out test split. The final results of the transfer experiments are reported in Table 3. The performed hyperparameter sweeps involved sweeping over the learning rates {.001, .01, 0.1, 0.2, 0.25, 0.3, 0.35, 0.4, 1., 2.}, batch sizes {128, 256, 512, 1024, 2048}, weight decay {1e−6, 1e−5, 1e−4, 1e−3, 0.01, 0.1}, warmup epochs {0, 10}, momentum {0.9, 0.99}, Nesterov {True, False}, and the number of training epochs. For the linear transfer protocol we considered setting epochs among {20, 30, 60, 80, 100}. Models were trained by stochastic gradient descent with momentum.

# B   INVARIANCE REGULARIZATION

We define the short-hands

$$p(\hat{z}_m^{a_1}; \tilde{z}_{m,t}^{a_2,+}) = \frac{\varphi(\hat{z}_m^{a_1}, \tilde{z}_{m,t}^{a_2,+})}{\varphi(\hat{z}_m^{a_1}, \tilde{z}_{m,t}^{a_2,+}) + \sum_{x_n \in \mathcal{N}(x_m)} \varphi(\hat{z}_m^{a_1}, \tilde{z}_{n,t}^{a_2})} \tag{5}$$

and

$$p(\hat{z}_m^{a_1}; \tilde{z}_{m,t}^{a_2}) = \frac{\varphi(\hat{z}_m^{a_1}, \tilde{z}_{m,t}^{a_2})}{\varphi(\hat{z}_m^{a_1}, \tilde{z}_{m,t}^{a_2}) + \sum_{x_n \in \mathcal{N}(x_m)} \varphi(\hat{z}_m^{a_1}, \tilde{z}_{n,t}^{a_2})} \tag{6}$$

where $\mathcal{N}(x_k)$ is the set of negatives, randomly uniformly sampled from the current batch, $\tilde{z}_{n,t}^{a_2} = g_t(f_t(x_n))$ is the target network projection of the negative sample; $\varphi(x_1, x_2) = \tau \cdot \exp(\langle x_1, x_2 \rangle / \tau)$ is the scoring function, $\tau > 0$ is a scalar temperature, and $\langle \cdot, \cdot \rangle$ denotes the standard Euclidean dot product.

We can now rewrite the components of the overall loss

$$\mathcal{L}_{\text{SEMPPL}} = \mathcal{L}_{\text{AUGM}} + \alpha \mathcal{L}_{\text{SEMPOS}} \tag{7}$$

as

$$\mathcal{L}_{\text{SEMPOS}} = -\sum_{m=1}^{B} \log p(\hat{z}_m^{a_1}; \tilde{z}_{m,t}^{a_2,+}) \tag{8}$$

and

$$\mathcal{L}_{\text{AUGM}} = -\sum_{m=1}^{B} \log p(\hat{z}_m^{a_1}; \tilde{z}_{m,t}^{a_2}). \tag{9}$$

As discussed in the main text we add the invariance penalty introduced in Mitrovic et al. [2021] to further increase the similarity between the anchor and positives and regularize the learning process. We add this invariance penalty both for augmentation positives and semantic positives. In particular, we compute

$$\begin{aligned} \mathcal{I}_{augm} &= D_{KL}(p(\hat{z}_m^{a_1}; \tilde{z}_{m,t}^{a_2}) \parallel p(\hat{z}_m^{a_2}; \tilde{z}_{m,t}^{a_1})) \\ &= sg[\mathbb{E}_{p(\hat{z}_m^{a_1}; \tilde{z}_{m,t}^{a_2})} \log p(\hat{z}_m^{a_1}; \tilde{z}_{m,t}^{a_2})] - \mathbb{E}_{p(\hat{z}_m^{a_1}; \tilde{z}_{m,t}^{a_2})} \log p(\hat{z}_m^{a_2}; \tilde{z}_{m,t}^{a_1}) \end{aligned}$$

and

$$\begin{aligned} \mathcal{I}_{sempos} &= D_{KL}(p(\hat{z}_m^{a_1}; \tilde{z}_{m,t}^{a_2,+}) \parallel p(\hat{z}_m^{a_2,+}; \tilde{z}_{m,t}^{a_1})) \\ &= sg[\mathbb{E}_{p(\hat{z}_m^{a_1}; \tilde{z}_{m,t}^{a_2,+})} \log p(\hat{z}_m^{a_1}; \tilde{z}_{m,t}^{a_2,+})] - \mathbb{E}_{p(\hat{z}_m^{a_1}; \tilde{z}_{m,t}^{a_2,+})} \log p(\hat{z}_m^{a_2,+}; \tilde{z}_{m,t}^{a_1}) \end{aligned}$$

where $sg$ denotes the stop-gradient operation. Taking all this together, this gives the final form of the loss as

$$\mathcal{L}_{\text{SEMPPL}} = c(\mathcal{L}_{\text{AUGM}} + \alpha \mathcal{L}_{\text{SEMPOS}}) + \lambda(\mathcal{I}_{augm} + \mathcal{I}_{sempos}) \tag{10}$$

with $\lambda$ the invariance scale and $c$ is the contrastive scale. We use $\lambda = 5$ and $c = 0.3$ in all our experiments irrespective of encoder size or training time as our method is robust to the choice of these hyperparameters.

## C PSEUDO-CODE OF SEMPPL

Listing 1 provides PyTorch-like pseudo-code for SEMPPL detailing how we compute pseudo-labels and use them to select the additional semantic positives, which are then used in the contrastive loss, along the augmentation positives.

```
1  '''
2  k: The number of neighbors in k-NN when computing pseudolabels.
3  f_o: online network: Encoder + comparison_net.
4  g_t: target network: Encoder + comparison_net.
5  gamma: Target EMA coefficient.
6  n_e: Number of negatives.
7  p_m: Mask apply probability .
8  '''
9  for i in range(num_large_views):
10     queue_i = queue.init(queue_size, FIFO)
11
12 for x, y in batch:  # Load batch of B samples each with data x and (maybe) label y.
13     x_m = mask_background(x)
14     for i in range(num_large_views):
15         # Stochastically apply background removal.
16         x = Bernoulli(p_m) ? x_m : x
17         # Create an augmented large view.
18         xl_i = augment(crop_large(x))
19         ol_i = f_o(xl_i)
20         tl_i = g_t(xl_i)
21         # Enqueue the labeled images in the batch.
22         if y is not None:
23             queue_i.enqueue((tl_i, y_i))
24
25     for i in range(num_small_views):
26         xs_i = augment(crop_small(x))
27         # Small views only go through the online network
28         os_i = f_o(xs_i)
29
30     # Pseudo-label computation for unlabelled examples.
31     if y is None:  # Missing label.
32         votes = [knn(k, queue_i, ol_j) for i, j in all_pairs(num_large_views)]
33         y = mode(votes)
34
35     loss = 0
36     # Compute the loss between all the pairs of large views.
37     for i in range(num_large_views):
38         for j in range(num_large_views):
39             loss += contrastive_loss(ol_i, tl_j, n_e)  # Augmentation positives.
40             for _ in range(num_semantic_positives):
41                 # Sample semantic positives from the queue, and add to the loss.
42                 z = sample(queue_j.filter(y))
43                 loss += contrastive_loss(ol_i, z, n_e)
44
45     # Compute the loss between the small and large views.
46     for i in range(num_small_views):
47         for j in range(num_large_views):
48             loss += contrastive_loss(os_i, tl_j, n_e)  # Augmentation positives.
49             for _ in range(num_semantic_positives):
50                 # Sample semantic positives from the queue, and add to the loss.
51                 z = sample(queue_j.filter(y))
52                 loss += contrastive_loss(ol_i, z, n_e)
53
54     loss /= ((num_large_views + num_small_views)
55     * num_large_views * (1 + num_semantic_positives))
56
57     # Compute the gradients, and update the online and target network.
58     loss.backward()
59     update(f_o)
60     g_t = gamma * g_t + (1 - gamma) * f_o
```

**Listing 1** Pseudo-code for SEMPPL.

# D  ANALYSIS

## D.1  IMPLEMENTATION DETAILS

We perform all the ablation experiments using $10\%$ of labelled data and train a standard ResNet-50 encoder with SEMPPL for 100 epochs (except in the training duration ablation). We report the top-1 accuracies on the ImageNet test set after fine-tuning from the first layer of the projector. As in Grill et al. [2020] and for the main results in this paper, we use multi-layer perceptrons for the projector and predictor with 2 linear layers—the first one followed by batch normalization [Ioffe & Szegedy, 2015] and rectified linear activation with output sizes 4096 and 256 for the two layers respectively. We use the same augmentations as for the experiments in the main paper—the standard SIMCLR augmentations [Chen et al., 2020a] and the RELICv2multi-crop and saliency-based masking [Tomasev et al., 2022]. Following the hyperparameter settings of the main results, we use

- batch size: $B = 4096$
- queue capacity $C = 20B$ (unless specifically ablated)
- number of nearest neighbours $k = 1$ (unless specifically ablated)
- view voting is used (unless specifically ablated)
- weight decay: $1e-6$ (exclude biases and batch normalization parameters)
- optimizer: LARS (exclude biases and batch normalization parameters from adaptation)
- base learning rate: 0.3 (scaled linearly with batch size [Goyal et al., 2017])
- warm-up: 10 epochs
- cosine decay schedule for learning rate
- exponential moving average parameter: 0.996
- views: 4 large views of size $224 \times 224$ and 2 small views of size $96 \times 96$
- temperature: $\tau = 0.2$
- number of semantic positives: 3 (unless specifically ablated)
- 10 randomly subsampled negatives per anchor
- $\alpha = 1/5$ (unless specifically ablated), $\lambda = 5$ and $c = 0.3$.

### D.2 ADDITIONAL ANALYSES

**Number of semantic positives** We study the effect of varying the number of semantic positives in SEMPPL. Table 8a shows that increasing this number from 1 to 3 only has an effect on the amount of correctly predicted pseudo-labels, but no effect on downstream performance. On the other hand, using 5 or 10 semantic positives significantly improves performance and also yields much more accurate pseudo-labels prediction.

**Training duration** Next, we vary the length of training representations and examine downstream performance. As can be seen from Table 8b, both the proportion of correctly predicted pseudo-labels and downstream performance improve with longer training up to 300 epochs but decrease if we continue training up to 500 epochs. This indicates with training longer than 300 epochs SEMPPL is starting to overfit, an observation consistent with phenomena reported elsewhere in the literature involving noisy labels [Li et al., 2019a; Kim et al., 2019; Liu et al., 2020; Han et al., 2018; Zhang & Sabuncu, 2018; Wang et al., 2019b].

Table 8: Top-1 accuracy (in %) on the ImageNet-1k test set, and accuracy (in %) of correctly predicted pseudo-labels at the end of training for semantic positives and training length experiments.

| Num. positives | Top-1 | Pseudo-label acc. |
|:---:|:---:|:---:|
| 1 | 69.9 | 68.6 |
| 2 | 69.9 | 70.9 |
| 3 | 69.9 | 69 |
| 5 | **71.0** | 72.8 |
| 10 | 70.7 | **72.9** |

| Training time (epochs) | Top-1 | Pseudo-label acc. |
|:---:|:---:|:---:|
| 100 | 69.9 | 69.2 |
| 200 | 72.4 | 76.8 |
| 300 | **72.7** | **77.9** |
| 500 | 72.3 | 75.6 |

(a) Varying the number of semantic positives.  (b) Varying the length of training.

**View voting** SEMPPL generates multiple views from a single instance image in order to learn representations. Those different views can be leveraged towards better pseudo-labels prediction. Rather than only picking one randomly selected data view to compute a single pseudo-label, we perform *majority voting* over (noisy) pseudo-labels computed from all available image views. Specifically, we compare the online predictor embedding of one view with the queue of the target projector embeddings of the same data view from previous batches in the first setting; in the second setting we compare the online predictor embedding of each view with the queue of the target projector embeddings of each other data view from previous batches.

Since SEMPPL relies on 4 large views, this yields up to 16 different pairs of views to compare and compute pseudo-labels from, i.e. we get 16 pseudo-label predictions; this setting we call *view voting*. Table 9 shows that using all available views to compute pseudo-labels significantly increases pseudo-labels accuracy which in turn significantly improves downstream performance.

Table 9: Top-1 accuracy (in %) on the ImageNet-1k test set, and accuracy (in %) of correctly predicted pseudo-labels at the end of training for view voting experiments (using all views to compute pseudo-labels vs using just a single view).

| View voting | Top-1 | Pseudo-label acc. |
|:---:|:---:|:---:|
| On | **69.9** | **68.6** |
| Off | 69.0 | 62.5 |

**Number of nearest neighbours** In order to compute pseudo-labels we use $k$-nearest neighbour lookup on the queue. While in the main results Section 3 we consistently assume $k = 1$ here we ablate the effect of varying $k$ on downstream performance. As can be seen Table 10a, increasing $k$ actually leads to a small *decrease* in performance. This is attributable to the decrease in the proportion of correctly predicted pseudo-labels as $k$ increases.

**Queue length**    How long should the queue of target projector embeddings for computing pseudo-labels be? As the queue grows in size, it contains increasingly stale embeddings and threatens to hamper the accuracy of predicted pseudo-labels. On the other hand, increasing queue size increases the amount and diversity of labelled embeddings available which we would expect to be beneficial. Those two opposing forces—staleness and increasing coverage—govern the accuracy with which we can correctly predict pseudo-labels in turn directly affecting the selection of semantic positives and their quality. We resolve this ambiguity empirically. As seen in Table 10b, increasing the coverage and diversity of labelled embeddings has a strong positive effect on representation learning and downstream performance. Staleness of embeddings is far less of a problem at practical (i.e., not very large) queue sizes, showing diversity and coverage to be the dominant factor.

Table 10: Top-1 accuracy (in %) on the ImageNet-1k test set, and accuracy (in %) of correctly predicted pseudo-labels at the end of training for $k$-NN and queue length experiments.

| **k**-nn | Top-1 | Pseudo-label acc. |
|---|---|---|
| 1 | **70.0** | **70.5** |
| 2 | 69.9 | 69.5 |
| 3 | 69.8 | 70.0 |
| 5 | 69.8 | 70.0 |
| 10 | 69.1 | 68.0 |

(a) Varying the number of nearest neighbours.

| Queue size (C) | Top-1 | Pseudo-label acc. |
|---|---|---|
| 4000 | 67.9 | 53.1 |
| 8000 | 68.6 | 60.4 |
| 12000 | 68.8 | 62.0 |
| 20000 | 69.2 | 64.3 |
| 40000 | 69.6 | 66.9 |
| 80000 | **69.9** | 68.8 |
| 200000 | 69.8 | **69.5** |

(b) Varying queue size.

**The effect of the loss weight $\alpha$**    We use $\alpha$ in Equation 4 to weight the contribution of the loss coming from semantic positives against the loss coming from augmentation positives. In SEMPPL we use multiple semantic positives for learning and thus we need to adjust $\alpha$ to appropriately weigh the contribution of the individual semantic positives. In our main experiments, we use $3$ semantic positives and for this reason $\alpha$ is not equal to $1$.

In Table 11 we vary $\alpha$ from 0.0 to 1.0, with $\alpha = 0.0$ effectively recovering RELICv2. The results indicate that the SEMPPL loss notably improves over RELICv2, but that the exact choice of $\alpha$ should be treated as a hyperparameter. Note that the value $0.2$ is very close to the $\frac{1}{3}$ which is exactly $1$/number of semantic positives. Thus, the optimal choice of $\alpha$ is very close to that fraction.

Table 11: Top-1 accuracy (in %) on the ImageNet-1k test set for varied values of the loss weight $\alpha$

| Loss weight ($\alpha$) | Top-1 |
|---|---|
| 0.0 (RELICv2) | 72.4 |
| 0.2 | **76.0** |
| 1.0 | 74.6 |

# E    AUGMENTATIONS

In this work, we follow the established data augmentations protocols and pipelines of Chen et al. [2020a]; Grill et al. [2020]; Caron et al. [2020]; Mitrovic et al. [2021]; Chen et al. [2020b]. Specifically, SEMPPL uses a set of augmentations to generate different views of the original image which has three channels, red $r$, green $g$ and blue $b$ with $r, g, b \in [0, 1]$.

The augmentations we use are generated by applying the following sequence of operations in the following order

1. Crop the image: Randomly select a patch of the image, between a minimum and maximum crop area of the image, with aspect ratio sampled log-uniformly in $[3/4, 4/3]$. Upscale the patch, via bicubic interpolation, to a square image of size $s \times s$.

2. Flip the image horizontally.

3. Colour jitter: randomly adjust brightness, contrast, saturation and hue of the image, in a random order, uniformly by a value in $[-a, a]$ where $a$ is the maximum adjustment (specified below).

4. Grayscale the image, such that the channels are combined into one channel with value $0.2989r + 0.5870g + 0.1140b$.

5. Randomly blur. Apply a $23 \times 23$ Gaussian kernel with standard deviation sampled uniformly in $[0.1, 2.0]$.

6. Randomly solarize: threshold each channel value such that all values less than $0.5$ are replaced by $0$ and all values above or equal to $0.5$ are replaced with $1$.

Apart from the initial step of image cropping, each subsequent step is applied with a certain probability to generate the augmented view of the original image. These probabilities and other augmentation parameters are given in Table 12. SEMPPL uses 4 large views of size $224 \times 224$ pixels and 2 small views of $96 \times 96$ pixels; to get the first and third large views and the first small view we use the parameters listed below for odd views, while for the second and fourth large view and the second small view we use the parameters for even views. Note that these are the same augmentations used also in Chen et al. [2020a]; Grill et al. [2020]; Caron et al. [2020]; Mitrovic et al. [2021]; Chen et al. [2020b].

In addition to these augmentations, we also randomly apply the saliency masking augmentation proposed in Tomasev et al. [2022] which enables us to remove a large part of the background. We follow the protocol described in Tomasev et al. [2022] for computing the saliency masking for an image and we apply this augmentation with probability $0.1$ to the 4 large views. In keeping with Tomasev et al. [2022], we fill out the removed background of the image with homogeneous grayscale noise with the grayscale level randomly sampled for each view. We only apply the saliency masking when the remaining foreground covers at least $20\%$ of the total image.

| Parameter | Even views | Odd views |
|---|---|---|
| Probability of randomly cropping | 50% | 50% |
| Probability of horizontal flip | 50% | 50% |
| Probability of colour jittering | 80% | 80% |
| Probability of grayscaling | 20% | 20% |
| Probability of blurring | 100% | 10% |
| Probability of solarization | 0% | 20% |
| Maximum adjustment $a$ of brightness | 0.4 | 0.4 |
| Maximum adjustment $a$ of contrast | 0.4 | 0.4 |
| Maximum adjustment $a$ of saturation | 0.2 | 0.2 |
| Maximum adjustment $a$ of hue | 0.1 | 0.1 |
| Crop size $s$ | 224 | 96 (small), 224 (large) |
| Crop minimum area | 8% | 5% (small), 14% (large) |
| Crop maximum area | 100% | 14% (small), 100% (large) |

Table 12: Parameters of data augmentation scheme. Small/large indicates small or large crop.

## F COMPUTATIONAL COST OF SEMPPL

**Added cost of SEMPPL**   As SEMPPL is based on RELICv2, the computational overhead of SEMPPL over RELICv2 comes from three factors: i) the queue maintenance, ii) the $k$-NN execution, and iii) the computation of the added loss term. We carefully analyzed this overhead, and concluded the following:

- The $k$-NN and the queue maintenance take $5.6\%$ of the step time, with the $k$-NN itself taking $4.5\%$.
- The total loss (both forward and backward passes of loss terms coming from both augmentation and semantic positives) takes $3.9\%$ of the step time, with $2.9\%$ belonging to the additional loss coming from semantic positives.
- In total, SEMPPL takes $8.5\%$ of the total step time.

Note that in our work we use a naive implementation of the $k$-NN. If needed, the $k$-NN can be further sped up, at a fraction of the accuracy, with a fast approximate $k$-NN model such as FAISS [Johnson et al., 2019]. This would further reduce the computation cost of using the $k$-NN to predict pseudo-labels and select semantic positives. Additionally, since our code is not heavily optimised, there is room for additionally lowering these numbers with targeted optimisations.

**Comparing SEMPPL to related models**   The direct comparison of training times of SEMPPL to related models can be deceiving, since there are three different sources of difference that influence the comparison. These summarise to differences in: i) accelerator architectures used, ii) deep learning frameworks used, and iii) concrete code, which can be assumed, is not optimised for any of the models, given their research nature. Being aware of these sources of difference, we contrast the running times of pre-training for SEMPPL and related models:

- PAWS [Assran et al., 2021] report $8.5$ hours per $100$ epochs on $64$ V100s. Assuming continuity in speed, this is equivalent to $25.5$ hours for $300$ epochs.
- SimMatch [Zheng et al., 2022] report $2.34$ hours per epoch on $8$ V100s. If we (generously) assume perfect scaling (which is difficult in reality) to $64$ V100s, this is equivalent to $87.75$ hrs for $300$ epochs.
- SEMPPL trains in $13$ hours on $64$ TPUv3 cores.

Given empirical evidence that TPUv3 is roughly $23\%$ faster than V100 on ResNet-50 training [Khairy, 2020], one could infer that the implementation of our model is significantly more computationally efficient. However, given that we cannot control all the aforementioned differences mentioned above, more careful analysis is needed to establish the right cost for each of these methods.

