# OpenReview forum: "SemPPL: Predicting Pseudo-Labels for Better Contrastive Representations"
_ICLR.cc/2023/Conference — ICLR 2023 poster_

### Official Review · Reviewer_JoKU · 2022-10-25

**Confidence:** 4
**Correctness:** 3
**Technical Novelty And Significance:** 3
**Empirical Novelty And Significance:** 4
**Recommendation:** 6

**Clarity, Quality, Novelty And Reproducibility:**

+ Clarity: The paper is in general well written.

+ Quality: Good performance on semi-supervised learning and transfer learning.

+ Novelty: The idea of combining augmentation-invariance and class-invariance from the labeled subset is simple but effective.

+ Reproducibility: The authors promise to open source code and model checkpoints on GitHub.

**Details Of Ethics Concerns:**

no.

**Strength And Weaknesses:**

### Pros:

+ The construction of semantic positives introduces additional class-invariant objective in addition to augmentation-invariant objective as in the instance-level contrastive training.

+ New state-of-the-art performance on semi-supervised learning with 1% and 10% ImageNet dataset.

### Cons:

- In SemPPL, the pseudo-labeling stage relies on a $k$-NN to retrieve semantically close labeled samples. What is the speed of $k$-NN compared with the rest of the training?

- What's the effect of loss weight$\alpha$? I am particularly interested in the case of $\alpha=0$, meaning that you drop the augmentation-invariant self-supervised objective and $\alpha=1$, meaning that you are reproducing a baseline from pure self-supervised representation (since I notice the augmentations used in SemPPL are stronger than the original SimCLR so the baseline might be higher than the reported number as well).

**Summary Of The Paper:**

The paper proposes to integrate the idea of pseudo-labeling to the contrastive self-supervised learning paradigm. To be specific, the method uses a subset of labeled data to pseudo-label the unlabeled data point through $k$-NN. The pseudo-labeled data can therefore be used as a semantically positive sample and joint optimized using the contrastive learning objective. The paradigm achieves outperforms previous methods by a noticeable margin on ImageNet with 1% or 10% data. It also shows strong performance on robustness, out-of-distribution and transfer learning settings.

**Summary Of The Review:**

Excellent performance and well-written technical contribution.

---

> ### Author Response · Authors · 2022-11-16
> **Response to reviewer JoKU**
>
> We thank the reviewer for their time and effort in reviewing our paper and for their insightful comments and suggestions on how to improve the paper.
>
> **Regarding the k-NN speed:**
>
> In our work we use a naive implementation of the k-NN. Running the k-NN algorithm takes 4.5% of the overall step time in training (5.6% when including the queue maintenance, for more details please take a look at our in-depth reply to reviewer iXGB). Note that the k-NN computations can easily be sped up, at a fraction of the accuracy, with a fast approximate k-NN model such as FAISS [1]. This would further reduce the computation cost of using the k-NN to predict pseudo-labels and select semantic positives.
>
> **Regarding the effect of loss weight α:**
>
> Note that we base SemPPL on the self-supervised contrastive method ReLICv2 [2]. We choose this method as this is currently the most performant self-supervised representation learning method. Thus, putting α=0 recovers the ReLICv2 method. Note that we use the same augmentations as ReLICv2 which extends the set of augmentations used in BYOL and SimCLR only by adding saliency masking. In the ReLICv2 paper, the authors show that the gain from saliency masking is rather small (+0.5% for linear evaluation on ImageNet).
>
> We use α to weight the contribution of the loss coming from semantic positives against the loss coming from augmentation positives. In SemPPL we use multiple semantic positives for learning and thus we need to adjust α to appropriately weight the contribution of the individual semantic positives. In our main experiments, we use 3 semantic positives and for this reason α is not equal to 1.
>
> Given α=0.0 recovers ReLICv2, and our reported results are for α=0.2, we additionally pretrained SemPPL for α=1.0 and evaluated it following the same evaluation protocol. The results, which we present in the following table, indicate that the SemPPL loss notably improves  over ReLICv2, but that the exact choice of α should be treated as a hyperparameter. Note that the value 0.2 is very close to the ⅓ which is exactly 1/number of semantic positives. Thus, the optimal choice of α is very close to 1/number of semantic positives.
>
> | Alpha         | Top-1 (%) |
> | ------------- | --------- |
> | 0.0 (ReLICv2) | 72.4      |
> | 0.2           | 76.0      |
> | 1.0           | 74.6      |
>
>
>
> [1] Johnson, J., Douze, M., & Jégou, H. (2019). Billion-scale similarity search with gpus. IEEE Transactions on Big Data, 7(3), 535-547.
>
> [2] Tomasev et al. “Pushing the limits of self-supervised ResNets: Can we outperform supervised learning without labels on ImageNet?”

---

> > ### Author Response · Authors · 2022-11-25
> > **Reply**
> >
> > Thank you very much for your review. We hope our reply has addressed your questions and comments. If there is anything else outstanding, please let us know. If we have satisfactorily addressed all your questions and concerns, we would kindly ask you to reevaluate your score in light of it.

---

> > > ### Comment · Reviewer_JoKU · 2022-12-07
> > > **Replying the authors' response**
> > >
> > > Dear Authors,
> > >
> > >  Thank you for the response. My concerns are now well addressed. I also see the clarifications get reflected in the updated draft. It looks better now.

---

> > > > ### Author Response · Authors · 2022-12-13
> > > > **Thank you**
> > > >
> > > > Thank you once again for your constructive proposals which have helped improve our work! We are very happy to have addressed your concerns and to have improved the paper based on your suggestions. We would be deeply grateful if you would consider reevaluating your score in light of these improvements.
> > > >
> > > > Thank you very much again for the great service of helping us improve our work!

---

### Official Review · Reviewer_vzyU · 2022-10-25

**Confidence:** 5
**Correctness:** 4
**Technical Novelty And Significance:** 3
**Empirical Novelty And Significance:** 4
**Recommendation:** 8

**Clarity, Quality, Novelty And Reproducibility:**

**Clarity/Quality**
Paper is written very well, it is simple to follow main ideas, results and settings. Several places can be improved further in the paper. Also some technical clarifications are needed in the text to improve the story and idea description.
- abstract: be consistent with specifying the results first for 1% then for 10% - otherwise hard to parse them.
- introduction, first paragraph, last two sentences: suggestion is to say about "labeled data could do earlier guidance and properly ground us from the beginning of training rather than only at the end at fine-tuning phase."
- introduce OOD notation before first usage in introduction.
- any links in introduction for positive mining? "virtuous cycle" - here good to reference to pseudo-labeling where one model is trained all the time and pseudo-labels continuously re-generated, e.g. fixmatch - it is the same story.
- nice to have small comment on why $l_2$ normalization is important (maybe it will work without it and just do normalization for KNN but not for overall training and SSL propagation through this normalization?)
- there is very limited description of queue / buffer in the main text. No any information about how we fill it at the beginning of training (if there no so labeled samples were seen yet. Do we train with SSL loss until we have filled the queue and then we start to do joint training? Do we have only labeled data in queue or we have both labeled and pseudo-labeled data later? All clarification are needed in the main text.
- Could authors add 1 sentence info about ablations in the main text: keep ablations in appendix, but add references into main text?
- Table 1 - not clear if simCLR has EMA in use. Did authors try other any SSL methods to see the generalization of the method for other SSL variants?
- Sec 3.2 paragraph before Table 3 - rewrite, seems there is some typos and not clear formulation of the content.
- Page 7 - precision-recall discussion please refer to Figure 2 somewhere in the text.
- what is an extra cost on top of SSL to do proposed positive mining? Should be negligible but wonder how it is comparable.
- Section 5 last paragraph on related works about transformer-specific trainings - one of the recent works in this direction is Zhai, S., et al., 2022, June. Position Prediction as an Effective Pretraining Strategy. In International Conference on Machine Learning (pp. 26010-26027). PMLR.
- After formula 10 why $\lambda$ and $c$ are not ones as there is phrase later "as our method is robust to the choice of hyperparameters"? I found weird to have 5 and 0.3 correspondingly for these params.
- Code listing, appendix C - queue_i var is not defined before.
- One suggestion about updating the queue is not to remove the oldest embedding, but rather sample randomly to decide which one to remove, or even to decide to we want to add new embedding into it randomly. This can balance the noise we have in pseudo-label and consistency with small changes in training.
- Some more advanced question: can we decrease batch size having now semantic positives? Do we still need such large batch?
- Table 10: not clear False and True notation here from the text. Maybe writing can be improved.
- Could authors confirm that in queue it is only labeled data representations and there could be one sample with several different representations obtained from different model snapshots for target branch?

**Novelty**
All pieces of the proposed algorithm appeared in prior literature, e.g. KNN, SSL, EMA, augmentations, additional MLP on top of representations, ensembling, pseudo-labeling, positive mining. However authors proposed a novel way of combining them and grounding SSL method on the use of labeled data by mining positive samples. The idea is new and interesting.

**Reproducibility**
A bunch of details are given throughout the paper and also in Appendix C, D.1, E, including the snippet of the code for algorithm, augmentation details and model main parameters.


**Details Of Ethics Concerns:**

No any concerns. The paper is about general algorithmic contribution.

**Strength And Weaknesses:**

**Strength**
- I like a new idea proposed in the paper in terms of how to perform positive mining and use some pseudo-labeling for this.
- Clear presentation and extensive ablations of the parameters like queue size, KNN size, OOD and domain transfer, large architecture, etc.
- SOTA results for 1/10% of supervision
- Analysis of pseudo-labels quality and how it changes over the training


**Weaknesses**
Part of weaknesses is just clarification into the main text which should be done to resolve some ambiguities. Others are some extra experiments to better understand the contribution on top of simCLR and strongly demonstrate that adding labeled data from the beginning helps SSL loss.
- there are no ablations on how penalty influences training and what is its contribution in vanilla simCLR v2
- Any thoughts / experiments if method is applicable to ViT? And how it is comparable with recent work of pseudo-labeling https://openreview.net/forum?id=7a2IgJ7V4W&referrer=%5BReviewer%20Console%5D(%2Fgroup%3Fid%3DNeurIPS.cc%2F2022%2FConference%2FReviewers%23assigned-papers)?
- It is not clear from the text if we do fine-tuning on labeled data of the full network or just linear probing. Thus not clear if the baselines are in the same setting as proposed method. Also not clear if simCLR v2 benchmark is done also with EMA for target branch of the SSL as it serves as the main purely comparable baseline having the same structure of the SSL except additional positive mining.
- I think it is worth to mention some limitations of the proposed method as overall discussion about pseudo-labeling could be misleading. Pseudo-labeling is applicable for **any** data type, including sequential like speech recognition and machine translation. Proposed method is not directly applicable to speech and text domain as there we operate with sequence level labels without segmentation.
- why did authors decide to use EMA? SimCLR v2 worked without it too. I wonder what is the impact of EMA overall on the method. It should work without it as authors maintain the queue of labeled data representations and there could be the same sample but with representations obtained from different model snapshots - kind of ensembling which performs same as EMA.
- there could be different ways (4 cases) to compute embedding for labeled / unlabeled data and then perform KNN. Why do authors select the way described in formula 2? any ablations or at least motivation (better to add it into the text)?
- Table 3, 8: I like results, this is great! But what are the results for simCLR here? Could it be that the method performs better because it is using SSL loss and not PAWS / other semi-sup. loss?
- Table 5: Why do we not get results of simCLR v2 from Table 1 for the last row in Table? What is the difference then as we removed both pseudo-labels and semantic positives? Also not clear if simCLR v2 baseline is also done with multi-crop strategy
- There is no any discussion on the proportion of labeled / unlabeled data in the main text and in the experiments? How do we balance data?


**Summary Of The Paper:**

There are two successful directions in DL nowadays to build models based on both labeled and large amount of unlabeled data: pseudo-labeling (semi-supervised) and SSL (self-supervised learning) with further fine-tuning. There are few papers which try to properly combine both SSL and supervised loss right from the beginning of training showing that maybe we should use labeled data earlier to ground ourselves or guide the training. Current paper is extending SSL with positive samples mining: we use supervised data to generate pseudo-labels based on KNN in representational space to find closest labeled data and use their label as pseudo-label. The pair is positive if their labels/pseudo-labels are the same. In this way SSL is grounded by some extra knowledge from labeled data. Also pseudo-labeling is used in non-standard way as we don't compute cross entropy on pseudo-labels but we use them to decide what is positive sample. Authors probe this idea with resnet model and 1/10% of labeled data from ImageNet. They achieve SOTA results with for resnet architecture. In extra ablations authors showed that the method works for large resnet-based models, for out-of-distribution generalization and transfer learning.

**Summary Of The Review:**

I think the main strong contribution of the paper is demonstration that self-supervised learning (SSL) (via successful and popular contrastive learning) can be significantly improved if supervised data are used not only at finetuning phase but also at pre-training phase, which is more aligned in the way how babies are learning with additional (weak) supervision and active learning. I hope this paper could push research community in the direction of SSL and semi-supervised learning synergy but not as two-phase training but more natural way, e.g. how this paper proposes.

Overall the idea is presented in a clear way and with extensive ablations. There are several concerns on the clear presentation and discussion of the queue component in the method as well as some extra ablations to strengthen the point "we should use labeled data as earlier as possible and integrate/ground SSL with them".


Update: Based on the revision and additional experiments and analysis I am changing my score from 6 to 8.

---

> ### Author Response · Authors · 2022-11-17
> **Response part 1**
>
> We sincerely thank the reviewer for their time and effort in reviewing our paper and for their many tremendously insightful comments and suggestions on how to improve the paper. We are deeply indebted for all the in-depth feedback and suggestions you provided us.
>
> **Re no ablations on how penalty influences training and what is its contribution in vanilla simCLR v2:**
>
> We implement SemPPL on top of the ReLICv2 method. We see where the confusion might stem from - the Table 1, where we omitted ReLICv2 results. Thank you for pointing that out, we’ll add ReLICv2 results to Table 1, and we’ll clarify (relating to your later question) that SemPPL without pseudo-labels and semantic positives in Table 5 is effectively ReLICv2.
>
> We additionally combined SemPPL with another performant self-supervised method, BYOL, and showed that SemPPL improves performance with BYOL as the base loss (please see table below). We will add these additional results into the main paper and also include further clarifications about our  method.
>
> | Method                      | 1%   | 10%  |
> | --------------------------- | ---- | ---- |
> | BYOL                        | 53.2 | 68.6 |
> | SemPPL on BYOL base loss    | 57.1 | 72.4 |
> |                             |      |      |
> | ReLICv2                     | 58.1 | 72.4 |
> | SemPPL on ReLICv2 base loss | 68.5 | 76.0 |
>
>
>
>
>
> **Re thoughts on and applicability to ViT:**
>
> Thank you for raising this interesting question!
> As no parts of our framework (pseudo-labelling and semantic positive selection) are architecture specific, we strongly believe that SemPPL would be applicable also to Vision Transformer backbones. This is  something that we hope to explore in future work.
>
> Comparison with the paper “Semi-supervised Vision Transformers at Scale” that you mention:
>
> a) Methodological comparison: While both papers use pseudo-labels, they are used in quite distinct ways that we believe them to actually be complementary and thus believe that SemPPL could be easily combined with their method. In particular, the authors of “Semi-supervised Vision Transformers at Scale” use pseudo-labels as targets within a cross-entropy loss after an initial self-supervised representation learning stage. On the other hand, we incorporate pseudo-labels for selecting more informative positives within the initial representation learning. Thus, their method could easily be extended with SemPPL in the pretraining stage. We base our belief on our additional experiments above, where we show that SemPPL works as an add-on to other SSL models.
>
> b) Experimental comparison: ResNet-50 and ViT are two different types of architectures with differing structure and number of parameters and require significantly different training protocols. As such, it is rather difficult to compare these models in a like-for-like manner as they are not directly comparable. Furthermore, from their results reported in their paper (Semi-supervised Vision Transformers at Scale) we see that SemPPL outperforms their method  on 1% labeled ImageNet data and that SemPPL performance is comparable on 10% labeled ImageNet data in the lower parameter regime though large ViTs are admittedly more performant.
>
> **Re fine-tuning on labelled data or linear probing:**
>
> We follow the fine-tuning protocol established by SimCLRv2 (Chen et al 2020b) and followed by subsequent work to ensure fair comparison. The fine-tuning is done only using the labelled data.
>
> **Re limitations:**
>
> SemPPL, in the exact form that is presented in the paper, will not work out of the box with sequences. As you pointed out, pseudo-labelling is applicable to any type of data. Thus, if we used e.g. a backbone which encodes a sequence into a single embedding, we could still still use the SemPPL method as presented - to calculate distances between (singleton) embeddings of sequences, pseudo-labelling them and selecting semantic positives based on that. On the other hand, if we have a sequence encoder which embeds an input sequence into a sequence of embeddings instead of a single embedding, we could imagine an approach that uses a combination of sequence alignment and distance calculation (e.g. dynamic time warping) that would produce a distance between sequences that we could then use for k-NN as currently used in SemPPL

---

> > ### Author Response · Authors · 2022-11-17
> > **Response part 2**
> >
> > **Re the use of EMA:**
> >
> > Please also note that we use ReLICv2 as the baseline method from which we build SemPPL.
> > We use the EMA network in SemPPL since ReLICv2 also uses the EMA network. The EMA network was introduced in BYOL and has been subsequently used in many self-supervised and semi-supervised methods (e.g. PAWS, the paper you suggested above “Semi-supervised Vision Transformers at Scale”).
> >
> > While SimCLRv2 does not use an EMA network, they use a memory buffer with moving average of weights for the selection of negatives (the authors mention this in point 3. in section 2 in the SimCLRv2 paper). Please note that this is very similar to using an EMA network; in the EMA network setup both positives and negatives come from a network with moving average weights instead of just the negatives as it is in SimCLRv2).
> >
> >
> > **Re 4 ways to compute embedding for labelled / unlabelled data:**
> >
> > We are unclear which 4 possible cases of computing embedding distances you refer to, so we’ll cover two possible interpretations.
> > 1) If you are referring to calculating distances between projections and predictions between the online and the target network, bear in mind that the target network only has a single MLP, thus it has only the projection embedding. We use the distance between the prediction of the online network and the projection of the target network because these two are directly optimised in the base loss.
> > 2) If you are referring to the fact that we are using 4 views in both the online network and the target network, we do use the all-to-all comparison between their embeddings, producing 16 k-NNs, and voting among them. We present this as view voting and show that it makes a noticeable difference in both the performance of the kNN and the Top1 accuracy.
> >
> >
> > **Re Omission of SimCLR in Table 3:**
> >
> > We omitted SimCLR from Table 3 for space reasons as it performs strictly worse than PAWS. We’re adding the comparison here and can expand the table in the paper.
> >
> > | Method | MF   | T-0.7 | Ti   | ImageNet-R | ObjectNet |
> > | ------ | ---- | ----- | ---- | ---------- | --------- |
> > | SimCLR | 53.2 | 61.7  | 68.0 | 18.3       | 14.6      |
> > |        |      |       |      |            |           |
> > | SemPPL | 65.4 | 74.1  | 79.6 | 24.4       | 25.3      |
> >
> >
> > **Re the omission of SimCLRv2 from Table 5:**
> >
> > Please note that SemPPL is based on the ReLICv2 loss. So when removing pseudo-labels and semantic positives, we get the ReLICv2 method.  However, you did uncover an unfortunate typo we made in rewriting our results, thank you. The 70.4% in that table should have been 72.4%. We will correct this and explicitly mark that SemPPL without pseudo-labels and semantic positives is equivalent to ReLICv2.
> >
> >
> > **Re discussion on the label disbalance:**
> >
> > Our understanding is that you are referring to the balance of ImageNet-1k, if it is not, please let us know. ImageNet-1k 1% and 10% are more-or-less balanced datasets, with on average 12.8 (12-13) and 128.1 (128-129) images per class. We did not test our model on datasets with class imbalance, though we hypothesize this could be achieved by balancing the queue, i.e. ensuring that the queue contains approximately similar number of embeddings per class. This would be interesting future work.

---

> > > ### Author Response · Authors · 2022-11-17
> > > **Response part 3**
> > >
> > > **Re clarity/quality comments:**
> > >
> > > We are very thankful for all the in-depth comments and suggestions you took time to give us; they’ve been a valuable source of feedback that will improve the clarity of our exposition. We will implement all the suggestions you made, and are further replying to specific questions made in the section:
> > >
> > > - **Re using SemPPL with other SSL methods** - we’ve added a table (above and also in reply to Reviewer U91Z) that shows how SemPPL can be combined with BYOL as a base loss too, outperforming the BYOL loss thus confirming SemPPLs compatibility with other SSL methods.
> > > - **Re the extra cost of positive mining:** we replied to Reviewer iXGB’s question on that in more detail, so please take a look at our detailed reply there. In short, SemPPL-specific operations take 8.5% of the total step time.
> > > - **Re $\lambda$ and $c$ in in Formula 10:** we took the values of these two hyperparameters from the list of the best ReLICv2 hyperparameters.
> > > - **Re queue updating:** you are right, there are multiple possible ways to decide which embeddings to add to/remove from the queue. From adding/replacing only high-precision (e.g. low-distance) ones, or ones with multiple supporting embeddings, to envisioning the queue as the way to balance heavy class imbalance (balance the queue so each class contains approx. the same number of instances on the queue). We particularly think class balancing as an interesting next step for the model
> > > - **Re decreasing the batch size:** this is a good question. We ran some quick preliminary experiments on the 10% dataset to quantify this. Originally, we got 76.0% for the batch size of 4096. We get a performance drop of 0.3% when using the batch size of 2048, and a performance drop of 1.5% when using the batch size of 1024. However, please note that we did not have the time to do an in-depth sweep over the learning rates, which should be done when changing the batch size, so it is quite possible that the model accuracy on lower batch sizes could be closer to our original experiments than the results we report here.
> > > - **Re contents of the queue:** yes, the queue contains only representations of labeled data. The chances of having one sample with several different representations are very small for the 10% labeled setting (due to queue size in relation to labelled dataset size). For the 1% labeled setting, there will be some samples that have different representations from different snapshots of the target network.

---

> > > > ### Comment · Reviewer_vzyU · 2022-11-18
> > > > **Thanks**
> > > >
> > > > Dear Authors,
> > > >
> > > > Thanks for detailed comments and additional experiments, now results look clear and strong for me!
> > > >
> > > > Best,
> > > > Reviewer.

---

> > > > > ### Comment · Reviewer_vzyU · 2022-11-23
> > > > > **Raising the score**
> > > > >
> > > > > Dear Authors,
> > > > >
> > > > > Thanks for updating the paper! I went through the last revision and it looks strong to me (especially having results on top of other SSL methods). I only suggest to include into final version the runtime evaluation you posted on KNN contribution to the total computations so that readers can know what to expect in practice.
> > > > >
> > > > > Based on the new revision and all answers I would like to raise my score to 8.
> > > > >
> > > > > Best,
> > > > > Reviewer.

---

> > > > > > ### Author Response · Authors · 2022-11-25
> > > > > > **Reply**
> > > > > >
> > > > > > Thank you very much for acknowledging our efforts, suggesting and recognising the improvements in the paper. We will add the runtime evaluation to the final version of the paper.

---

### Official Review · Reviewer_iXGB · 2022-11-03

**Confidence:** 3
**Correctness:** 3
**Technical Novelty And Significance:** 2
**Empirical Novelty And Significance:** 2
**Recommendation:** 6

**Clarity, Quality, Novelty And Reproducibility:**

The clarity of the paper can be improved. As mentioned above in Cons, there are several confusing descriptions.

The novelty of the proposed method is marginal. Using the nearest labeled datapoints in the embedding space as pseudo-labels of unlabeled datapoints is a common practice in the semi-supervised setting.

Most of the details are included in the paper, so the proposed method should be reproducible.

**Strength And Weaknesses:**

Pros:

- Interesting topic. It is interesting and important to utilize the supervised information in semi-supervised setting rather than a simple cross-entropy objective. This paper proposes to use that information to help decide the similarity relationships in the embedding space for contrastive learning.
- Good experiments. The proposed method outperforms state-of-the-art semi-supervised learning methods on ImageNet with 1% and 10% labels. The analytical and supplemental experiments are also comprehensive.

Cons:

- The proposed method is naive. It just uses the nearest labeled datapoint as the pseudo-label of each unlabeled datapoint and then adds a supervised contrastive loss using the labels and the pseudo-labels directly. Moreover, the description is confusing. For example, what’s the purpose of first retrieving the k-nearest neighbors and then finding the nearest one from the k datapoints (Equation 2)? In 2.1 Algorithm parameters, what does it mean by |a| = 4 (what is a?) and why are there 16 pseudo-labels for each unlabeled datapoint?

- The diversity of the datasets used is too low. It compared with previous methods for semi-supervised learning only on ImageNet, which makes the empirical evaluation less convincing. Most of the baselines included in the tables of this paper (e.g., DebiasPL [Wang et al., 2022], SimMatch [Zheng et al., 2022], etc) present the results on other datasets (e.g., CIFAR) in addition to ImageNet. This is no reason to exclude those results and compare the results on only one dataset.

- This is no computation cost analysis in the paper. Since the proposed method has much more steps than the traditional contrastive learning framework, it would be helpful to evaluate how much computation overhead it brings. Comparison to other semi-supervised methods (e.g., as SimMatch [Zheng et al., 2022] does) is also necessary.

**Summary Of The Paper:**

This paper extends contrastive learning to semi-supervised settings. To do so, it estimates pseudo-labels for the unlabeled data during the training process of contrastive learning and adds a supervised contrastive loss according to the labels of the labeled data as well as the pseudo-labels of the unlabelled data to the original contrastive loss. Experiments demonstrate that it achieves a new state-of-the-art performance in semi-supervised benchmarks and has better robustness and generalizability.

**Summary Of The Review:**

This paper addresses an interesting topic, i.e., how to utilize the supervised information in the semi-supervised setting to help contrastive learning to learn a better embedding space. However, as discussed above, the proposed method lacks novelty and some experiments are missing. Therefore, I feel this paper is below the acceptance threshold.

---

> ### Author Response · Authors · 2022-11-16
> **Response part 1**
>
> We thank the reviewer for their time and effort in reviewing our paper and for their insightful comments and suggestions on how to improve the paper.
>
> **Re method naivete:**
>
> In our work we purposefully focus on proposing simple, but novel and highly performant methods. We believe that simplicity is a virtue in scientific endeavors as it promotes better understanding, facilitates reproducibility and dissemination of ideas and methods.
>
> While pseudo-labeling and semantic positives might seem as simple, straightforward ideas, to the best of our knowledge, these ideas have not been utilized together in previous work and we are the first to propose to combine them, leading to our novel highly performant semi-supervised method SemPPL. Note that we have made a considerable effort to correctly quantify the contribution of the different model elements (ablations) and performance with respect to other models (comparison to state-of-the-art competing methods).
>
> We believe that our novel approach, although it might be perceived as simple in nature, is a valuable contribution to the machine learning community especially since it achieves state-of-the-art results in multiple benchmarks and outperforms more complex methods, e.g. SimMatch.
>
> **Re confusing description:**
>
> Thank you for pointing this out and we’re happy to clarify our text further. Concretely, we calculate cosine similarities between all the embeddings in the queue with the projection of the anchor. We then choose the nearest one (1-NN) in all of our reported experiments, except when ablating $k$ where we use the most occurring class (effectively picking the mode) of the top k nearest neighbours. We will update the text accordingly.
>
> As for the significance of $a$: $a$ are the augmentations produced, meaning we produce $4$ augmentations per anchor. We keep all $4$ augmentations (views) in the queue, and build a k-NN for each pair of (anchor augmentation, augmentation saved in the queue), thus resulting in $16$ comparisons in total (effectively making 16 k-NNs). We call this view voting and we get 16 votes since we have 16 k-NNs. We use majority voting to decide which label will be assigned to the anchor. We perform ablations on the effect of view voting (Appendix D) and present a precision and recall analysis. We will update the text in 2.1 accordingly.
>
> **Re low diversity of datasets:**
>
> We focused on ImageNet as it is a large-scale dataset and is by far the most commonly used dataset in the literature for pretraining. CIFAR-10 and CIFAR-100 are far less commonly used in the literature as pretraining datasets as they are significantly smaller than ImageNet and thus learning general representations with these datasets is more challenging and subsequently these representations are less useful for transfer. To put things into perspective, CIFAR-10 and CIFAR-100 contain 50k images of 32x32 resolution, spanning over 10 and 100 classes, respectively, whereas ImageNet-1k contains 1.2M images of 224x224 resolution with 1000 classes.
>
> While we pretrain only on ImageNet, note that we test the learned representations on a large number of widely different datasets (both in-distribution and out-of-distribution to ImageNet) to showcase the generality of our method and the usefulness of the learned representation. Taking together all of our experiments, we test the learned SemPPL representations on 16 datasets in addition to ImageNet (c.f. transfer and robustness/ood results). Among these 16 datasets we also include CIFAR-10 and CIFAR-100. Please also note that we test the downstream performance of SemPPL on more datasets than competing methods SimMatch and PAWS did.

---

> > ### Author Response · Authors · 2022-11-16
> > **Response part 2**
> >
> > **Re computation cost analysis:**
> >
> > Thank you for raising this point. First, we’d like to state that our method does not have much more steps above the base model (which is ReLICv2 in our case). The additional steps are the queue maintenance, knn execution and the computation of the added loss term. We carefully analyzed the overhead of SemPPL on top of ReLICv2, and concluded the following:
> > The kNN and the queue maintenance take 5.6% of the step time (the kNN itself taking 4.5%)
> > The total loss (both forward and backward pass of loss terms coming from both augmentation and semantic positives) takes 3.9% of the step time, with 2.9% belonging to the additional loss coming from semantic positives
> > In total, SemPPL takes 8.5% of the total step time
> > Note that:
> > kNN can be sped up, at a fraction of the accuracy, with a fast approximate kNN model such as FAISS [1]
> > Our code is not heavily optimised so there is a good chance that these numbers could easily be additionally lowered
> >
> > Second, the direct comparison of training times of SemPPL to related models can be deceiving, since there are three different sources of difference that influence the comparison: i) different accelerator architectures used, ii) different deep learning frameworks used, and lastly, iii) different code, which we assume, is not optimised in any case, given the research nature of all these experiments. Given these rather large grains of salt, we contrast the running times of pretraining for the competing methods:
> > PAWS reports 8.5 hrs per 100 epochs on 64 V100s. Assuming continuity in speed, this is equivalent to 25.5 hours for 300 epochs
> > SimMatch reports 2.34 hrs per epoch on 8 V100s. If we generously assume perfect scaling (which is difficult in reality) to 64 V100s, that is equivalent to 87.75 hrs for 300 epochs
> > SemPPL trains in 13 hrs on 64 TPUv3 cores.
> >
> > Since there is evidence that TPUv3 is 23% faster than V100 on ResNet-50 training [2], one could easily say the implementation of our model is significantly more computationally efficient.
> > Given we cannot control all the aforementioned differences, we restrict ourselves from making that claim.
> >
> > We are happy to add this discussion in the appendix of the paper.
> >
> > [1] Johnson, J., Douze, M., & Jégou, H. (2019). Billion-scale similarity search with GPUs. IEEE Transactions on Big Data, 7(3), 535-547.
> > [2] TPU vs GPU vs Cerebras vs Graphcore: A Fair Comparison between ML Hardware https://khairy2011.medium.com/tpu-vs-gpu-vs-cerebras-vs-graphcore-a-fair-comparison-between-ml-hardware-3f5a19d89e38

---

> > > ### Author Response · Authors · 2022-11-25
> > > **Reply**
> > >
> > > Thank you very much for your review. We've further improved the clarity of the paper per your suggestions, and we hope to have addressed your questions, comments and concerns. Please let us know if there are any further questions or comments. If we have addressed all your questions and concerns satisfactorily, we would kindly ask to reevaluate your score in light of this.

---

> > > > ### Comment · Reviewer_iXGB · 2022-12-06
> > > > **Response**
> > > >
> > > > Thanks for the response. My concerns are mostly addressed. Although I am still a bit concerned about the novelty of the method, based on its performance, I am willing to raise my score.

---

> > > > > ### Author Response · Authors · 2022-12-13
> > > > > **Thank you**
> > > > >
> > > > > Dear Reviewer,
> > > > >
> > > > > thank you again for your time and consideration of our work and the thoughtful questions, comments and discussion!
> > > > > Thank you also very much for raising your score!

---

### Official Review · Reviewer_15Rd · 2022-11-03

**Confidence:** 4
**Clarity, Quality, Novelty And Reproducibility:** 1) My major concern is that the novel…
**Correctness:** 3
**Technical Novelty And Significance:** 2
**Empirical Novelty And Significance:** Not applicable
**Recommendation:** 6

**Strength And Weaknesses:**

1) The idea is straightforward and sound.
2) The experiments are sufficient to demonstrate its effectiveness.
3) paper is written well and easy to follow.

**Summary Of The Paper:**

This paper adapts the contrastive learning from self-supervised learning to semi-supervised learning with two core designs: 1) performing pseudo-labeling based on the similarities to the labeled data in the encoded feature space; 2) select positives for a datapoint based on the pseudo-labels.

Pros: The idea is simple and extensive experiments demonstrate its effectiveness.
Cons: The novelty of the proposed method is relatively limited.


**Summary Of The Review:**

While the idea is simple yet effective and the experiments are extensive, the novelty is limited, not adequate for ICLR.

---

> ### Author Response · Authors · 2022-11-17
> **Response to reviewer**
>
> We thank the reviewer for their time and effort in reviewing our paper and for their insightful comments and suggestions on how to improve the paper.
>
> Our proposed method SemPPL incorporates learning representations with predicting pseudo-labels and selecting semantic positives based on pseudo-labels into a single representation learning framework. As already pointed out by the reviewer, this framework is a novel contribution to the literature. To the best of our knowledge, pseudo-labelling and semantic positives have not previously been studied jointly, and one of the key contributions of our paper is showing that combining those two simple ideas is very powerful (as demonstrated by our extensive experimentation and state-of-the-art results) and underexplored. To the best of our knowledge,  we would also like to draw the reviewer’s attention to the fact that our paper is the first work to use predicted pseudo-labels to select semantic positives for contrastive learning.
>
> In our paper, we purposefully propose a simple, yet effective method for semi-supervised learning with limited amounts of labeled data. The simplicity of our method enables others to easily reimplement it and facilitates the reproducibility of our experimental results.
> Furthermore, the simplicity of SemPPL makes this method general and widely applicable, e.g. we could easily combine SemPPL with other self-supervised objectives, such as BYOL (which we do in the reply to reviewer U91Z; we reproduce it also below for completeness). We also hypothesize we can easily use SemPPL in conjunction with different architectures, e.g. Vision Transformers.
>
> | Method                      | 1%   | 10%  |
> | --------------------------- | ---- | ---- |
> | BYOL                        | 53.2 | 68.6 |
> | SemPPL on BYOL base loss    | 57.1 | 72.4 |
> |                             |      |      |
> | ReLICv2                     | 58.1 | 72.4 |
> | SemPPL on ReLICv2 base loss | 68.5 | 76.0 |
>
> In summary, our proposed method SemPPL is simple, general, performant, and easy to implement while outperforming the previous state of the art. We therefore believe that SemPPL represents a valuable contribution to semi-supervised learning and can be of significant interest to the machine learning community in general.

---

> > ### Comment · Reviewer_15Rd · 2022-12-06
> > **Response addresses most of my concerns**
> >
> > Thanks for the concrete response.
> >
> > The explanation  in the response convinces me of higher evaluation of the novelty. Indeed, it is the first time to combine two simple ideas in Contrastive Learning: 1) learning pseudo-labels and 2) selecting semantic positives. More importantly, the added experiments further validates the effectiveness of the proposed simple method. Thus, I am willing raise my score from 5 to 6.

---

> > > ### Author Response · Authors · 2022-12-13
> > > **Thank you**
> > >
> > > Dear reviewer,
> > >
> > > thank you for your time and effort in examining our paper and the additional experiments.
> > > Thank you also very much for raising your score!

---

> ### Author Response · Authors · 2022-11-28
> **Thank you**
>
> Thank you for your time and consideration of our work and for your review. We hope that we have addressed your concerns with our response. Please do not hesitate to let us know if there are any outstanding questions or comments.
>
> We would also kindly ask you to consider your evaluation in light of our response and consider raising your score. Thank you very much!

---

### Official Review · Reviewer_U91Z · 2022-11-04

**Confidence:** 4
**Correctness:** 4
**Technical Novelty And Significance:** 2
**Empirical Novelty And Significance:** 3
**Recommendation:** 6

**Clarity, Quality, Novelty And Reproducibility:**

The paper has written in a clear and precise manner.

While this idea of combining the pseudo labels while learning visual representations during contrastive learning has been introduced in SimMatch (Zheng et al, 2022), the method discussed in this paper is more elegant and original.

**Strength And Weaknesses:**

**Strengths**
_S1_. The paper provides an elegant solution to combine pseudo labels (semantic similarity) and instance similarity while learning visual representations. This approach is more elegant than SimMatch (Zheng et al, 2022), beating their performance with lesser number of training epochs.
_S2_. It achieves state of the art on 1% and 10% ImageNet settings using ResNet-50 architecture.
_S3_. The paper beats the transfer learning performance w.r.t. SimMatch (Zheng et al, 2022).


**Weaknesses**
As such the paper is well written with a good set of ablation studies. But certain things aren't clear.
_W1_. The paper says that is uses the Relic objective (Mitrovich et. al., 2021). It is not clear how important this objective would be in terms of the performance gain. This objective could be used for the competing approaches as well.
_W2_. It would be good to have the precision and recall plots of the pseudo label quality (Fig. 2) for the competing approaches. That would give good insight into whether there is a correlation with this elegant way of optimizing to the quality of pseudo labels generated. It would also justify the point the authors are marking about the virtuous cycle of learning better representations while improving quality of pseudo labels. Currently, that point has not been justified empirically. We know the approach helps learning better representations.
_W3_. It would be good to also show how this approach does on limited annotation settings introduced in PAWS (Assran et. al., 2021) and Masked Siamese Networks (Assran et. al., ECCV 2022) with 1-10 annotations per class.

**Summary Of The Paper:**

The paper proposes an elegant way of combining pseudo labels (semantic similarity) and instance similarity. This method achieves state of the art performance on 1% and 10% ImageNet settings using ResNet-50 architecture. Transfer learning and OOD generalization experiments show the usefulness of this approach.

**Summary Of The Review:**

The paper provides an elegant solution to the idea introduced by SimMatch (Zheng et. al. 2022). It is well written and there is extensive evaluation done w.r.t. semi-supervised learning approaches in the literature, along with transfer learning and OOD generalization benchmarks. The novelty of the idea is not large. But it is an interesting contribution to the literature.

---

> ### Author Response · Authors · 2022-11-17
> **Response to reviewer**
>
> We thank the reviewer for their time and effort in reviewing our paper and for their insightful comments and suggestions on how to improve the paper.
>
> **Re W1; use of ReLIC objective:**
>
> We focus in our work on the ReLICv2 objective as it is the most performant self-supervised objective in terms of learning general and transferable representations [1]. Based on your suggestion, we ran experiments that combine another performant, self-supervised objective, namely BYOL [2] with SemPPL to show the generality of our idea on both contrastive and non-contrastive methods. Importantly, we follow the training pipeline (e.g. augmentation, hyperparameters etc.) from the original papers for each method (ReLICv2 and BYOL) to isolate the impact of SemPPL, and thus avoid confounding factors. In the below table, we report the top-1 accuracy on the ImageNet test set after pretraining and fine-tuning with 1% and 10% labeled ImageNet data. From the table below, we see that SemPPL has a positive effect on both base objectives.
>
> | Method                      | 1%   | 10%  |
> | --------------------------- | ---- | ---- |
> | BYOL                        | 53.2 | 68.6 |
> | SemPPL on BYOL base loss    | 57.1 | 72.4 |
> |                             |      |      |
> | ReLICv2                     | 58.1 | 72.4 |
> | SemPPL on ReLICv2 base loss | 68.5 | 76.0 |
>
>
> **Re W2; precision and recall:**
>
> Note that our precision and recall analysis depends on having an ensemble of k-NNs for computing pseudo-labels (in our case 16). As such, this analysis is not applicable to competing methods as they either do not use pseudo-labels at all (e.g. SimCLRv2) or they have a single pseudo-label predictor which could only give us precision but not recall. In addition, due to different training regimes of competing methods, even the precision analysis would not be directly comparable.
>
> We think a comparison that would give a better insight into the relationship between learning better representations and improving the quality of pseudo labels would be the comparison that contrasts the precision and recall as presented in Figure 2 with the precision and recall while i) turning off the use of pseudo-labels and ii) turning off the use of semantic positives altogether (even for the ground-truth labeled data), as that would enable us to track the same precision/recall values while opting not to use pseudo-labels for representation learning. We are running these two experiments and will report the results as soon as they’re done.
>
>
> **Re W3; Regarding 1-10 annotations per class:**
>
> Note that using 1% labeled ImageNet data to pretrain corresponds to 12-13 examples per class which is very close to the upper bound of examples that the reviewer was interested in.
>
> PAWS (Assran et al. 2021) proposes a semi-supervised method for ResNets and learns representations using 1% and 10% labeled ImageNet data, but does not address the 1-10 annotations per class setting in their paper. PAWS performs an ablation of the size of their support set (and they comment on how many images per class are in the support set) while still using 1% or 10% labeled data in training. On the other hand, Masked Siamese Networks (MSN) (Assran et al. 2022) uses Vision Transformers to learn representations and does explore how their method performs with 1, 2 and 5 images per class.
>
> In our work, we focus on ResNets and 1% and 10% labeled datapoints as these are standardized settings for which the appropriate datasets can be used by anyone (https://www.tensorflow.org/datasets/catalog/imagenet2012_subset). Thus, all the competing  methods using 1% and/or 10% labeled data report performance on exactly the same datasets enabling a fair comparison. Furthermore, the 1% and 10% labeled data settings are also the most commonly used settings in the recent literature combining self-supervised and semi-supervised learning.
> Consequently we compare against PAWS as this represents the most fair comparison. On the other hand, the 1-10 examples per class datasets are not standardised, meaning that when used, they are all generated on the fly hence these experimental results are not directly comparable as different subsets of the full dataset can be selected.
>
> [1] Tomasev et al. “Pushing the limits of self-supervised ResNets: Can we outperform supervised learning without labels on ImageNet?”
>
> [2] Grill et al. “Bootstrap your own latent: A new approach to self-supervised learning”

---

> ### Author Response · Authors · 2022-11-28
> **Precision and Recall Analysis**
>
> Per our previous reply, we ran additional experiments to assess the virtuous cycle of representation learning and pseudo-label prediction  You can find the figures for comparison here:
>
> - SemPPL (original figure from paper): https://ibb.co/3kBfzPh
> - SemPPL without pseudo-labels (SemPPL w/o PL): https://ibb.co/kJQZbns
> - SemPPL without semantic positives altogether (SemPPL w/o SP): https://ibb.co/jLY4hw0
>
> There is a stark difference between SemPPL and SemPPL w/0 PL, and  SemPPL and SemPPL w/0 SP. In Table 4 in the main paper, we see that SemPPL achieves a higher top-1 accuracy than SemPPL w/o PL and SemPPL w/o SP. Now, we also see (from the figures above) that there is a significant increase in both precision and recall of SemPPL compared to SemPPL w/o PL and SemPPL w/o SP. In particular, for the lowest precision threshold, there is a jump in precision from 0.44 to 0.68, and the recall for the highest threshold jumps from 0.2 to 0.47. We can also systematically observe consistent improvements across all the thresholds for both the precision and the recall. Finally, we see that this consistency in improvement is valid from the beginning to the end of training. This demonstrates the virtuous cycle of learning better representations (higher accuracy achieved) while improving the quality of pseudo-labels (higher precision and recall), compared to the cases when pseudo-labels are not used.
>
> Note that the difference in precision and recall between SemPPL w/o PL and SemPPL w/o SP is small as neither are using pseudo-labels for representation learning and only a small fraction of semantic positives are based on ground-truth labels.

---

> ### Author Response · Authors · 2022-11-28
> **Thank you**
>
> Thank you for your time and consideration of our work, and thank you for your review and the questions and comments.
> We hope to have addressed your concerns and questions with our detailed response and analysis.
> Please do not hesitate to let us know if there are any outstanding questions or comments.
>
> We would also kindly ask you to consider your evaluation in light of our response and consider raising your score. Thank you very much!

---

### Decision · Program_Chairs · 2023-01-20

**Decision:**

Accept: poster

**Justification For Why Not Higher Score:**

- Lack of technically novel components

**Justification For Why Not Lower Score:**

- Conceptually novel and interesting idea of semi-supervised representation learning
- Strong empirical results and analyses

**Metareview: Summary, Strengths And Weaknesses:**

This paper proposes using pseudolabels to improve representations learnt by contrastive learning in a semi-supervised representation learning setup, which is conceptually interesting even though the method itself is a combination of well-studied components. Improvements are demonstrated through downstream evaluations on semi-supervised learning, robustness and out-of-distribution generalization tasks. Experiments were extensive in terms of downstream tasks and contrastive losses, though largely limited to ResNet-style backbones and representations learnt using ImageNet.

**Note From Pc:**

if the above contains the word "oral" or "spotlight" please see: "oral" presentation means -> notable-top-5% and "spotlight" means -> notable-top-25%. As stated in our emails, we are disassociating presentation type from AC recommendations

**Summary Of Ac-Reviewer Meeting:**

Reviewers appreciated the conceptual contribution of using pseudolabelling for representation learning though several had concerns that the proposed method involved a combination of well-known ideas and thus lacked technical novelty. Reviewers also appreciated that the proposed method was simple and straightforward and showed strong performance in extensive experiments. The ablation studies and additional analyses provided (some after the response period) strengthened the reviewers' confidence in the effectiveness of the method. There were initial concerns about the fairness of comparisons (due to differing amounts of augmentation used) and computational cost but these were resolved in the author response. Overall, all reviewers concurred that the paper should be accepted, though majority did not recommend a higher score due to novelty concerns stated above.

The AC agrees that the conceptual contribution is interesting and supported by strong empirical studies, and that it should be accepted as one of the first works proposing this concept of semi-supervised contrastive representation learning.